# Deciphering regulatory architectures of bacterial promoters from synthetic expression patterns

**Rosalind Wenshan Pan**[1]*, **Tom Röschinger**[1], **Kian Faizi**[1], **Hernan G. Garcia**[2,3,4,5,6], **Rob Phillips**[1,7]*

1 Division of Biology and Biological Engineering, California Institute of Technology, Pasadena, California, United States of America, 2 Biophysics Graduate Group, University of California, Berkeley, California, United States of America, 3 Department of Physics, University of California, Berkeley, California, United States of America, 4 Department of Molecular and Cell Biology, University of California, Berkeley, California, United States of America, 5 Institute for Quantitative Biosciences-QB3, University of California, Berkeley, California, United States of America, 6 Chan Zuckerberg Biohub-San Francisco, San Francisco, California, United States of America, 7 Division of Physics, Mathematics, and Astronomy, California Institute of Technology, Pasadena, California, United States of America

* rosalind@caltech.edu (RWP); phillips@pboc.caltech.edu (RP)

**Data Availability Statement:** All code used in the work are available open source at https://github.com/RPGroup-PBoC/theoretical_regseq.

## Abstract

For the vast majority of genes in sequenced genomes, there is limited understanding of how they are regulated. Without such knowledge, it is not possible to perform a quantitative theory-experiment dialogue on how such genes give rise to physiological and evolutionary adaptation. One category of high-throughput experiments used to understand the sequence-phenotype relationship of the transcriptome is massively parallel reporter assays (MPRAs). However, to improve the versatility and scalability of MPRAs, we need a "theory of the experiment" to help us better understand the impact of various biological and experimental parameters on the interpretation of experimental data. These parameters include binding site copy number, where a large number of specific binding sites may titrate away transcription factors, as well as the presence of overlapping binding sites, which may affect analysis of the degree of mutual dependence between mutations in the regulatory region and expression levels. To that end, in this paper we create tens of thousands of synthetic gene expression outputs for bacterial promoters using both equilibrium and out-of-equilibrium models. These models make it possible to imitate the summary statistics (information footprints and expression shift matrices) used to characterize the output of MPRAs and thus to infer the underlying regulatory architecture. Specifically, we use a more refined implementation of the so-called thermodynamic models in which the binding energies of each sequence variant are derived from energy matrices. Our simulations reveal important effects of the parameters on MPRA data and we demonstrate our ability to optimize MPRA experimental designs with the goal of generating thermodynamic models of the transcriptome with base-pair specificity. Further, this approach makes it possible to carefully examine the mapping between mutations in binding sites and their corresponding expression profiles, a tool useful not only for developing a theory of transcription, but also for exploring regulatory evolution.

**Funding:** T.R. was supported by Boehringer Ingelheim Fonds. R.P. acknowledges support from the NIH (award number 1R35 GM118043). H.G.G. was supported by NIH R01 Awards R01GM139913 and R01GM152815, by the Koret-UC Berkeley-Tel Aviv University Initiative in Computational Biology and Bioinformatics, and by a Winkler Scholar Faculty Award. H.G.G. is also a Chan Zuckerberg Biohub Investigator (Biohub-San Francisco). The content is solely the responsibility of the authors and does not necessarily represent the official views of the NIH. The funders had no role in study design, data collection and analysis, decision to publish, or preparation of the manuscript.

**Competing interests:** The authors have declared that no competing interests exist.

## Author summary

With the rapid advancement of sequencing technology, there has been an exponential increase in the amount of genomic sequence data available from diverse organisms. Nevertheless, deciphering the genotype-phenotype mapping of this data remains a formidable task, especially when dealing with non-coding sequences such as the promoter. In current databases, annotations for transcription factor binding sites are sorely lacking, which creates a challenge for developing a systematic theory of transcriptional regulation. To address this gap in knowledge, high-throughput methods such as massively parallel reporter assays (MPRAs) have been employed to decipher the regulatory genome. In this work, we make use of thermodynamic models to computationally simulate MPRAs in the context of transcriptional regulation and produce thousands of synthetic MPRA datasets for bacterial promoters. We examine how well typical experimental and data analysis procedures of MPRAs are able to recover common regulatory architectures under different sets of experimental and biological parameters. By establishing a dialogue between high-throughput experiments and a physical theory of transcription, our efforts serve to both improve current experimental procedures and enhancing our broader understanding of the sequence-function landscape of regulatory sequences.

## 1 Introduction

With the widespread emergence of sequencing technology, we have seen an explosion of genomic data in recent years. However, data on transcriptional regulation remains far behind. Even for organisms as widely studied as *E. coli*, many promoters lack annotations on the transcription factor binding sites that underlie transcriptional regulation. Moreover, existing binding site annotations are largely without experimental validation for functional activity, as a large proportion are determined through DNA-protein interaction assays such as ChIP-Seq [1–4] or computational prediction [5]. This fundamental gap in knowledge poses a major obstacle for us to understand the spatial and temporal control of cellular activity, as well as how cells and organisms respond both physiologically and evolutionarily to environmental signals.

One strategy to understand the regulatory genome is by conducting massively parallel reporter assays (MPRAs), where the regulatory activities of a library of sequences are measured simultaneously via a reporter. The library of sequences may be genomic fragments [6] or sequence variants containing mutations relative to the wild-type regulatory sequence [7]. There are two main ways to measure regulatory activities in MPRAs. The first approach uses fluorescence-activated cell sorting to sort cells into bins based on the expression levels of a fluorescent reporter gene [8]. Subsequently, deep sequencing is utilized to determine which sequence variant is sorted into which bin. The second approach uses RNA-sequencing (RNA-Seq) to measure the counts of barcodes associated with each sequence variant as a quantitative read-out for expression levels. The two approaches have been used in both prokaryotic [9–12] and eukaryotic systems [13–15] to study diverse genomic elements including promoters and enhancers.

In particular, our group has developed Reg-Seq [16], an RNA-Seq-based MPRA that was used to successfully decipher the regulatory architecture of 100 promoters in *E. coli*, with the hope now to complete the regulatory annotation of entire bacterial genomes. Mutations in regulatory elements lead to reduced transcription factor binding, which may result in measurable changes in expression. Therefore, the key strategy to annotate transcription factor binding

sites based on MPRA data that we focus on is to identify sites where mutations have a high impact on expression levels. To do this, one approach is to calculate the mutual information between base identity and expression levels at each site. A so-called information footprint can then be generated by plotting the mutual information at each position along the promoter. Positions with high mutual information are identified as putative transcription factor binding sites.

In this paper, we make use of a new generation of thermodynamic models to simulate the RNA-Seq-based MPRA. Specifically, we make use of equilibrium statistical mechanics to build synthetic datasets that simulate the experimental MPRA data and examine how various parameters affect the output of MPRAs. Conventionally, thermodynamic models are not sequence specific. The binding energies are usually phenomenological parameters that are fit once and for all. By way of contrast, the new version of thermodynamic model leverages experimentally determined or synthetically engineered energy matrices which allow us to consider arbitrary binding site sequences and to compute their corresponding level of expression. Here, we primarily focus on bacterial promoters; however, given the prevalent use of thermodynamic models in model organisms such as *Drosophila* and mammalian cell lines [17–28], we are confident that the general approach presented in this work can be adapted to study eukaryotic systems as well.

Using a computational pipeline, we examine tens of thousands of unique promoters and hence tens of thousands of unique implementations of the sequence-specific thermodynamic models. These sequences are then converted into the two primary summary statistics used to analyze the experimental data, namely, information footprints and expression shift matrices. Given that the experimental implementations of these MPRAs entail tens of millions of unique DNA constructs, the sequence-specific thermodynamic models give us the opportunity to systematically and rigorously analyze the connection between key parameters. These parameters include experimental parameters, such as the rate of mutation used to generate the sequence variants, as well as biological parameters such as transcription factor copy number, where a large number of specific binding sites may titrate away transcription factors. These analyses will help us to optimize MPRA experimental design with the goal of accurately annotating transcription factor binding sites in regulatory elements, while revealing the limits of MPRA experiments in elucidating complex regulatory architectures. Additionally, the insights gained from our simulation platform will enable further dialogue between theory and experiments in the field of transcription, including efforts to understand how mutations in the evolutionary context give rise to altered gene expression profiles and resulting organismal fitness.

Our use of thermodynamic models is motivated by several principal considerations. First, because of their simplicity, these models have served and continue to serve as a powerful null model when considering signaling, regulation, and physiology. Their application runs the gamut from the oxygen binding properties of hemoglobin [29–33], to the functioning of membrane-bound receptors in chemotaxis and quorum sensing [34–37], and to the binding of transcription factors at their target DNA sequences [38–46]. Second, for our purposes, thermodynamic models form an internally consistent closed theoretical system in which we can generate tens of thousands of "single-cell" expression profiles and use the same tools that we use to evaluate real MPRA data to evaluate these synthetic datasets, thus permitting a rigorous means for understanding real data. That said, despite their many and varied successes, thermodynamic models deserve continued intense scrutiny since many processes of the central dogma involve energy consumption and thus may involve steady state probabilities that while having the form of ratios of polynomials, are no longer of the Boltzmann form [18, 40, 47–49]. These shortcomings of thermodynamic models are further discussed in S1 Appendix.

To address these shortcomings, we also provide a preliminary study of how broken detailed balance might change the interpretation of summary statistics such as information footprints.

The remainder of this paper is organized as follows. In Sec 2.1.1, we introduce our procedure to construct and analyze synthetic datasets for both promoters regulated by a single transcription factor and promoters regulated by a combination of multiple transcription factors. In Sec 2.1.2 and Sec 2.1.3, we discuss the choice of parameters related to the construction of the mutant library, including the rate of mutation, mutational biases, and library size. After setting up the computational pipeline, we perturb biological parameters and examine how these perturbations affect our interpretation of MPRA summary statistics. The parameters that we will explore include the free energy of transcription factor binding (Sec 2.2.1), the regulatory logic of the promoter (Sec 2.2.2), the copy number of the transcription factor binding sites (Sec 2.2.3), and the concentration of inducers (Sec 2.2.4). Next, we explore factors that may affect signal-to-noise ratio in information footprints. These factors include stochastic fluctuations of transcription factor copy number (Sec 2.3.1), non-specific binding events along the promoter (Sec 2.3.2), as well as the presence of overlapping binding sites (Sec 2.3.3). Additionally, in Sec 2.3.4, we generalize our pipeline and consider the cases of transcriptional regulation where detailed balance may be broken. Finally, we discuss the insights generated from our results in relation to future efforts to decipher regulatory architectures in diverse genomes.

## 2 Results

### 2.1 Mapping sequence specificity and expression levels

**2.1.1 Computational pipeline for deciphering regulatory architectures from first principles.** In MPRAs, the goal is to make the connection between regulatory sequences, transcription factor binding events, and expression levels. In Reg-Seq for example, the authors start with a library of sequence variants for an unannotated promoter, each of which contains a random set of mutations relative to the wild type sequence. Then, RNA-Seq is used to measure the expression levels of a reporter gene directly downstream of each promoter. By calculating the mutual information between mutations and the measured expression levels, the regulatory architecture of the promoter can be inferred. Finally, Bayesian inference methods can be used in conjunction with thermodynamic models to infer the interaction energies between transcription factors and their binding sites in absolute $k_B T$ units at base-by-base resolution [16].

Our computational MPRA pipeline involves similar steps, but instead of starting from experimental measurements of expression levels, we use thermodynamic models to predict expression levels given the sequences of the promoter variants and the corresponding interaction energies, as schematized in Fig 1. Through this process, we generate synthetic datasets of expression levels that are in the same format as the datasets obtained via RNA-Seq. Subsequently, we produce summary statistics from the synthetic datasets in the same way as we would produce summary statistics from an experimental dataset. Importantly, we can perturb various experimental and biological parameters of interest and examine how changing these parameters affect our ability to discover unknown transcription factor binding sites through MPRAs.

We first demonstrate the use of our sequence-specific thermodynamic models using a promoter with the simple repression regulatory architecture, i.e. the gene is under the regulation of a single repressor. Specifically, we use the promoter sequence of lacZYA. We assume that it is transcribed by the $\sigma^{70}$ RNAP and only regulated by the lactose repressor (LacI), which binds to the O1 operator within the lacZYA promoter. In S2 Appendix, we derive the probability of RNAP being bound ($p_{\text{bound}}$) for a gene with the simple repression regulatory architecture

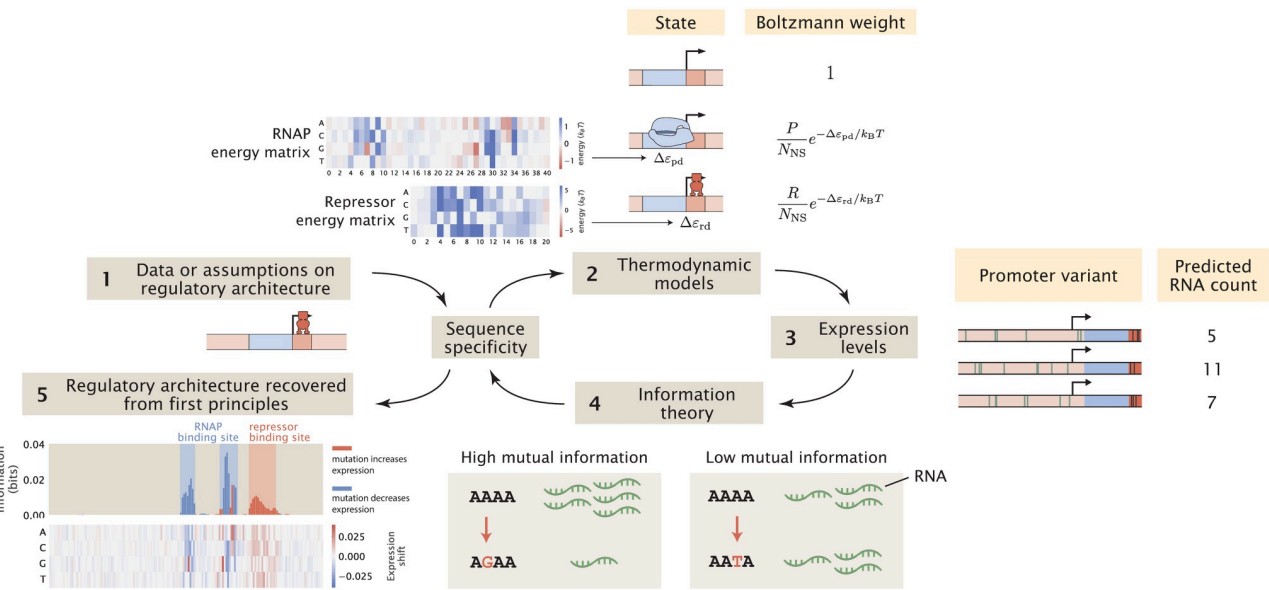

**Fig 1. A computational pipeline for deciphering regulatory architectures from first principles.** Given (1) knowledge or assumptions about the regulatory architecture of a promoter, we make use of (2) thermodynamic models to construct a states-and-weights diagram, which contains information about all possible states of binding and the associated Boltzmann weights. Here, in the states-and-weights diagram, $P$ is the copy number of RNAP, $R$ is the copy number of the repressor, $N_{NS}$ is the number of non-specific binding sites, and $\Delta\varepsilon_{pd}$ and $\Delta\varepsilon_{rd}$ represent the binding energies of RNAP and the repressors at their specific binding sites relative to the non-specific background, respectively. Using these states-and-weights diagrams as well as the energy matrices, which are normalized to show the change in binding energies for any mutation along the promoter compared to the wild-type sequence, we can (3) predict the expression levels for each of the promoter variants in a mutant library. To recover the regulatory architecture, we (4) calculate the mutual information between the predicted expression levels and mutations at each position along the promoter according to Eq 6. In particular, there is high mutual information if a mutation leads to a large change in expression and there is low mutual information if a mutation does not lead to a significant change in expression. The mutual information at each position is plotted in an information footprint, where the height of the peaks corresponds to the magnitude of mutual information, defined in Eq 6, and the peaks are colored based on the sign of expression shift, defined in Eq 9. Given the assumption that the positions with high mutual information are likely to be RNAP and transcription factor binding sites, we (5) recover the regulatory architecture of the promoter. The base-specific effects of mutations on expression levels can also be seen from expression shift matrices, which are calculated using Eq 10, where the difference between the expression levels of sequences carrying a specific mutation at a given position and the average expression level across all mutant sequences is computed.

[50, 51]. The final expression is given by

$$p_{bound} = \frac{\frac{P}{N_{NS}} e^{-\beta\Delta\varepsilon_{pd}}}{1 + \frac{P}{N_{NS}} e^{-\beta\Delta\varepsilon_{pd}} + \frac{R}{N_{NS}} e^{-\beta\Delta\varepsilon_{rd}}}. \tag{1}$$

Here, $\beta = \frac{1}{k_B T}$ where $k_B$ is Boltzmann's constant and $T$ is temperature. As illustrated in Fig 2, the parameters that we need are the number of non-specific binding sites ($N_{NS}$), the copy number of RNAP ($P$), the copy number of the repressor ($R$), and the binding energies for RNAP and the repressors ($\Delta\varepsilon_{pd}$ and $\Delta\varepsilon_{rd}$). We begin by assuming that $P$ and $R$ are constant and $N_{NS}$ is the total number of base pairs in the *E. coli* genome. On the other hand, the values for $\Delta\varepsilon_{pd}$ and $\Delta\varepsilon_{rd}$ depend on the sequence of the promoter variant.

We calculate the binding energies by mapping the sequences of the promoter variants to the energy matrices of the RNAP and the repressor, as shown in Fig 3A. Specifically, we assume that binding energies are additive. This means that given a sequence of length $l$, the

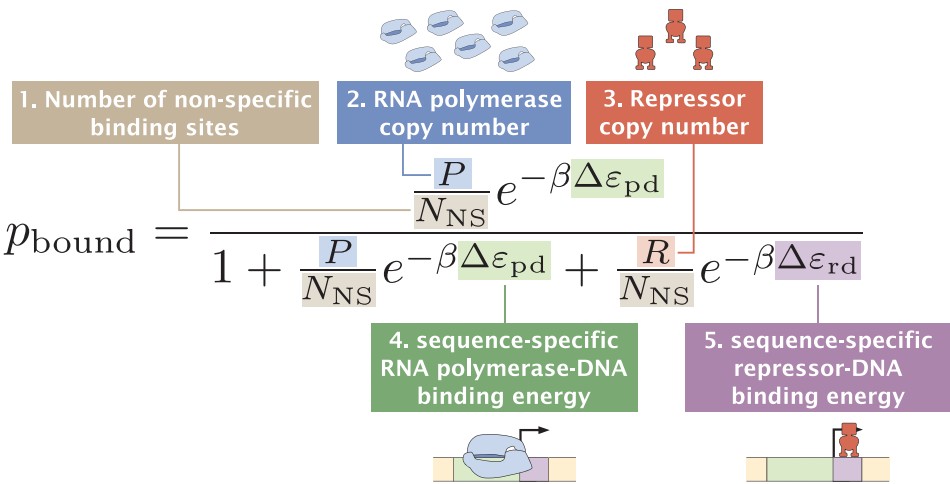

**Fig 2. Required parameters for the thermodynamic model of a promoter with the simple repression regulatory architecture.** For a promoter with the simple repression regulatory architecture, five parameters are required. These parameters are (1) the number of non-specific binding sites, $N_{\mathrm{NS}}$, (2) the copy number of the RNA polymerase, $P$, (3) the copy number of the repressor, $R$, (4) the sequence-specific binding energy of the RNA polymerase at its binding site, $\Delta\varepsilon_{\mathrm{pd}}$, and (5) the sequence-specific binding energy of the repressor at its binding site, $\Delta\varepsilon_{\mathrm{rd}}$.

total binding energy $\Delta\varepsilon$ can be written as

$$\Delta\varepsilon = \sum_{i=1}^{l} \varepsilon_{i,b_i}, \tag{2}$$

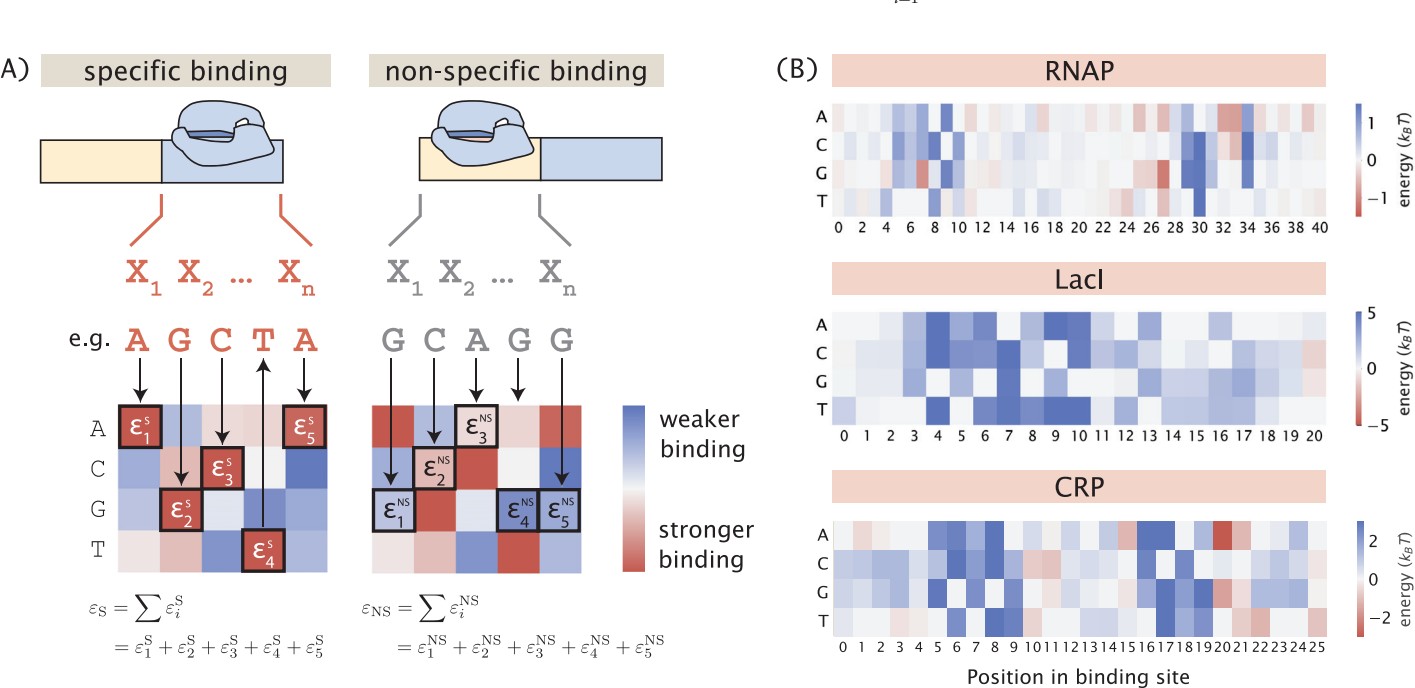

**Fig 3. Mapping binding site sequences to binding energies using energy matrices.** (A) Given the assumption that binding energies are additive, we can use an energy matrix to determine how much energy each base along the binding site contributes and compute the total binding energy by taking the sum of the binding energies contributed by each position. The total binding energy can be used to compute the Boltzmann weight for each of the states, which is then used to calculate the probability of RNAP being bound. (B) Experimentally measured energy matrices of RNAP [52], the repressor LacI [53], and the activator CRP [8].

where $\varepsilon_{i,b_i}$ is the binding energy corresponding to base identity $b_i$ at position $i$ according to the energy matrix. Here, we use the energy matrices of the RNAP and LacI that were previously experimentally determined using Sort-Seq [52, 53], which are shown in Fig 3B. Unless otherwise specified, these energy matrices are used to build all synthetic datasets in the remainder of this paper. It should be acknowledged that the additive model does not take into account epistasis effects [54, 55]. It has been shown that a linear model is largely sufficient to describe protein-DNA interactions in many cases [56–58]. Nevertheless, it may be beneficial to include higher-order interaction energy terms in future simulations of MPRAs.

After computing the sequence-specific binding energies, we can substitute the relevant energy terms into Eq 1 and calculate $p_\text{bound}$. To connect $p_\text{bound}$ to expression levels, we make use of the occupancy hypothesis, which states that the rate of mRNA production is proportional to the probability of RNAP occupancy at the promoter [59]. The rate of change in mRNA copy number is given by the difference between the rates of mRNA production and degradation. In general, there can be multiple transcriptionally active states, each with its own transcription rate. For example, for a promoter that is regulated by an activator, there are two transcriptionally active states, one where only RNAP is bound to the promoter, and one where both RNAP and the activator are bound to the promoter. Each of these two states could have a different rate of mRNA production. With this, we define an average rate of mRNA production, which is given by the sum of each state's production rate, weighted by the probability of the state $\langle r \rangle = \Sigma_i r_i\, p_{\text{bound},i}$. Hence, the rate of change of mRNA copy number is given by

$$\frac{\mathrm{d}m}{\mathrm{d}t} = \sum_i r_i\, p_{\text{bound},i} - \gamma m, \tag{3}$$

where for transcriptionally active state $i$, $r_i$ is the rate of transcription, $p_{\text{bound},i}$ is the probability of RNAP occupancy in state $i$, $m$ is the copy number of mRNAs, and $\gamma$ is the rate of mRNA degradation. Therefore, the steady-state level of mRNA is given by

$$m^* = \frac{1}{\gamma} \sum_i r_i\, p_{\text{bound},i}. \tag{4}$$

For simplicity, we assume that each transcriptionally active state has the same rate of mRNA production, $r$. Therefore,

$$m^* = \alpha \sum_i p_{\text{bound},i}, \quad \text{where } \alpha = \frac{r}{\gamma}. \tag{5}$$

Using the above expression, we can calculate the expected RNA count for each of the promoter variants in our library. Assuming that $r$ and $\gamma$ do not depend on the mRNA sequence, the total probability of RNAP being bound, given by $p_\text{bound} = \Sigma p_{\text{bound},i}$, is scaled by the same constant to produce the mRNA count of each promoter variant. Therefore, the probability distribution of expression levels is not affected by the choice of $\alpha$. Depending on sequencing depth, the mRNA count for each promoter variant is typically on the order of 10 to $10^3$ [16]. Here, we take the geometric mean and set $\alpha$ to $10^2$ to ensure that the mRNA count is on a realistic scale.

Up until this point, we have constructed a synthetic RNA-Seq dataset containing the predicted expression levels of each sequence variant in a mutant library. MPRA data is often described using several summary statistics. Using thousands of synthetically derived mRNA counts, we can compute such summary statistics and ask how they are altered by biological and experimental parameters. These summary statistics can then be used to infer the underlying regulatory architecture from these large-scale synthetic datasets. One of these summary

statistics is called an information footprint, which shows the mutual information between mutations and expression levels at each position along the promoter relative to the transcription start site (TSS). The mutual information at position $i$ is given by

$$I_i = \sum_b \sum_\mu \Pr_i(b,\mu)\log_2\left(\frac{\Pr_i(b,\mu)}{\Pr_i(b)\Pr(\mu)}\right),$$  (6)

where $b$ represents base identity, $\mu$ represents expression level, $\Pr_i(b)$ is the marginal probability distribution of mutations at position $i$, $\Pr(\mu)$ is the marginal probability distribution of expression levels across all promoter variants, and $\Pr_i(b,\mu)$ is the joint probability distribution between expression levels and mutations at position $i$. In general, $b$ can be any of the four nucleotides, i.e. $b \in \{A, C, G, T\}$. This means that $\Pr_i(b)$ is obtained by computing the frequency of each base per position. Alternatively, a more coarse grained approach can be taken, where the only distinction is between the wild-type base and mutation, in which case $b$ is defined as

$$b = \begin{cases} 0, & \text{if the base is mutated,} \\ 1, & \text{if the base is wild type.} \end{cases}$$  (7)

In S3 Appendix, we compare the signal-to-noise ratio of the information footprint when the coarse-grained definition of $b$ is used and when all four letters are used. We see that using the coarse-grained definition of $b$ improves the signal-to-noise ratio of the information footprint as reducing the number of states reduces artificial noise outside of the specific binding sites. Therefore, we use this definition for our subsequent analysis.

On the other hand, to represent expression levels as a probability distribution, we group sequences in each range of expression levels into discrete bins and compute the probabilities that a given promoter variant is found in each bin. Again in S3 Appendix, we calculate the signal-to-noise ratio of the information footprint as a function of the number of bins, and we find that increasing the number of bins leads to a lower signal-to-noise ratio in the information footprints because the additional bins contribute to artificial noise. Therefore, we choose to use only two bins with the mean expression level as the threshold between them. This means that $\mu$ can take the values of

$$\mu = \begin{cases} 0, & \text{if expression is lower than mean expression} \\ 1, & \text{if expression is higher than mean expression.} \end{cases}$$  (8)

In S3 Appendix, we derive the information footprint for a constitutive promoter analytically and demonstrate that in the absence of noise, mutual information is expected to be 0 outside of the specific binding sites and non-zero at a specific binding site.

To decipher the regulatory architecture of a promoter, another important piece of information is the direction in which a mutation changes expression. This can be determined by calculating another summary statistic called the expression shift, which measures the change in expression level when there is a mutation at each given position along the promoter [60]. Suppose there are $n$ promoter variants in our library, then the expression shift $\Delta s_l$ at position $l$ is given by

$$\Delta s_l = \frac{1}{n}\sum_{i=1}^n \xi_{i,l}(c_i - \langle c \rangle), \text{ where } \langle c \rangle = \frac{1}{n}\sum_{i=1}^n c_i,$$  (9)

where $c_i$ represents the RNA count of the $i$-th promoter variant, $\xi_{i,l} = 0$ if the base at position $l$

in the $i$-th promoter variant is wild type, and $\xi_{i,l} = 1$ if the base is mutated. If the expression shift is positive, it indicates that mutations lead to an increase in expression and the site is likely to be bound by a repressor. On the other hand, a negative expression shift indicates that mutations lead to a decrease in expression, and therefore the site is likely to be bound by RNAP or an activator.

To have a more precise understanding of how much each mutation to each possible base identity $b \in \{A, C, G, T\}$ changes expression levels, we can extend Eq 9 to calculate an expression shift matrix. Specifically, the value in the expression shift matrix at position $l$ corresponding to base $b$ is given by

$$\Delta s_{b,l} = \begin{cases} \dfrac{1}{n}\sum_{i=1}^{n}\xi_{b,i,l}\left(\dfrac{c_i}{\langle c \rangle} - 1\right), & \text{where } \langle c \rangle = \dfrac{1}{n}\sum_{i=1}^{n}c_i, & \text{if } b \text{ is mutated,} \\ 0, & & \text{if } b \text{ is wildtype} \end{cases} \tag{10}$$

where $\xi_{b,i,l} = 1$ if the base at position $l$ in the $i$-th promoter variant corresponds to base identity $b$ and $\xi_{b,i,l} = 0$ otherwise. Note that in comparison to Eq 9, here we are calculating the relative change in expression, which is easier to interpret than the absolute change in expression. An example of an expression shift matrix is shown in panel (5) of Fig 1.

In Fig 4B, we plot the information footprint and expression shift matrix for a promoter with the simple repression regulatory architecture. There are two peaks with negative expression shifts near the -10 and -35 positions, which correspond to the canonical RNAP binding sites. There is another peak immediately downstream from the transcription start site with a positive expression shift, which corresponds to the binding site of LacI. Taken together, we have demonstrated that by calculating mutual information and expression shift, we are able to recover binding sites from our synthetic dataset on expression levels.

Using the procedure described above, we can also produce synthetic datasets for other classes of regulatory architectures. In Fig 4, we demonstrate that we can recover the expected binding sites based on synthetic datasets for six common types of regulatory architectures [16]. The states-and-weights diagrams and $p_{\text{bound}}$ expressions used to produce the synthetic datasets for each of the architecture are shown in S3 Appendix. In addition, as demonstrated in S3 Appendix, similar sets of signals are observed in the information footprints generated from our synthetic datasets as those generated from experimental Reg-Seq data for the same architectures [16].

**2.1.2 Changing mutation rates and adding mutational biases.**   One key parameter in MPRAs is the level of mutation for each sequence variant in the library. Here, we again consider a gene with the simple repression regulatory architecture as a case study and we examine how varying mutation rates and mutational biases changes the signals in the information footprints. We quantify the level of signal, $S$, by calculating the average mutual information at each of the binding sites. This is given by

$$S = \langle I \rangle_B = \frac{1}{l}\sum_{i \in B}I_i, \tag{11}$$

where $B$ represents the set of bases within a given binding site, $I_i$ represents the mutual information at base position $i$, and $l$ is the length of $B$, i.e. the number of bases in the binding site.

As shown in Fig 5A and 5B, in general, when there is a higher rate of mutation, the average mutual information at the RNAP binding site increases relative to the average mutual information at the repressor binding site. To explain this effect, we consider $\kappa$, the ratio between the

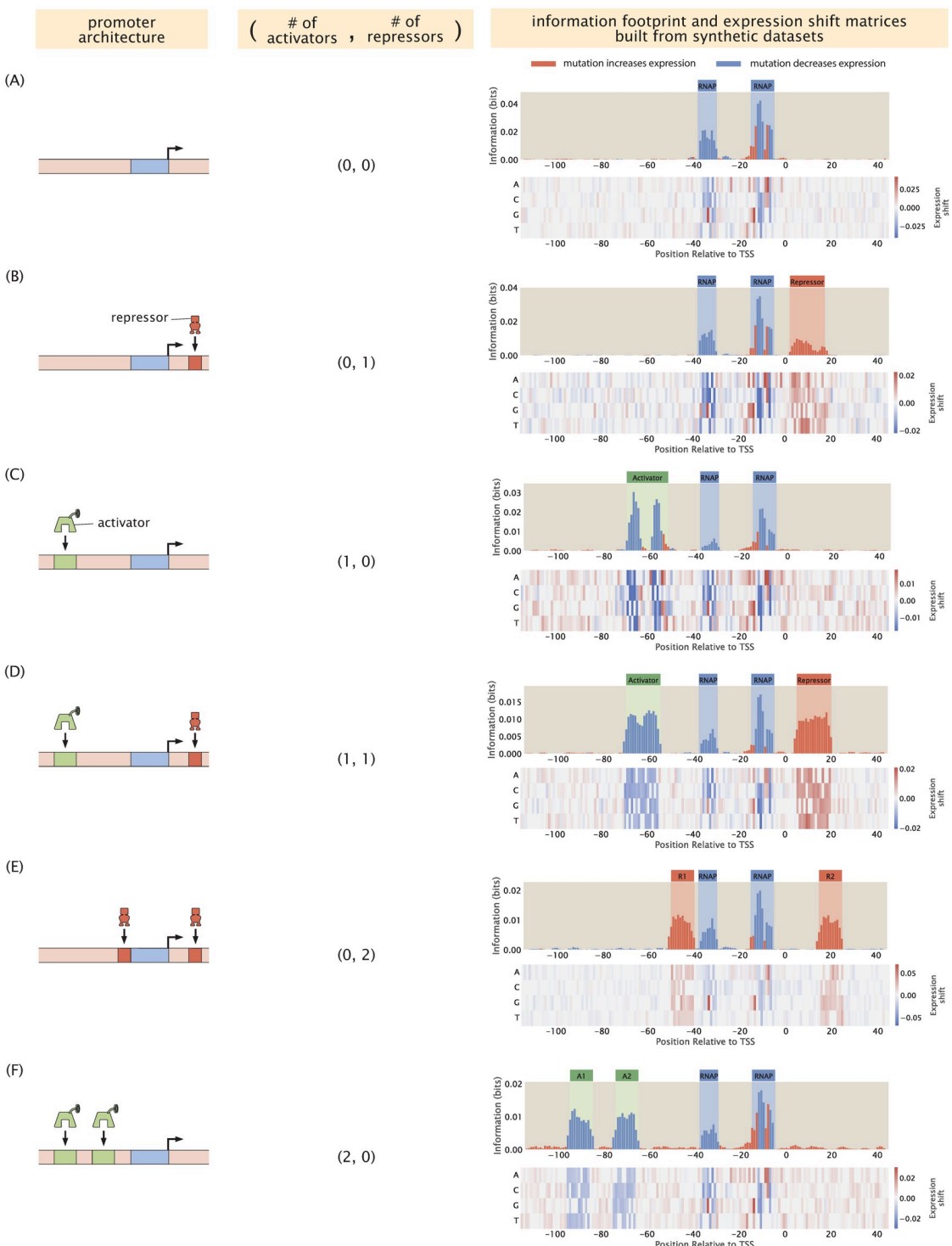

**Fig 4. Building information footprints and expression shift matrices based on synthetic datasets of different regulatory architectures.** We describe each of the regulatory architectures using the notation (A,R), where A refers to the number of activator binding sites and R refers to the number of repressor binding sites. The corresponding information footprints and expression shift matrices built from synthetic datasets are shown on the right. The architectures shown in panels A-F are constitutive expression, simple repression, simple activation, repression-activation, double repression, and double activation, respectively. For panels A-C, we use energy matrices of RNAP, LacI, and CRP shown in Fig 3B. For panels D-F, we continue to use the experimentally measured energy matrix for RNAP; the energy matrices for the repressors and the activators are constructed by hand, where the interaction energies at the wild type bases are set to 0 $k_B T$ and the interaction energies at the mutant bases are set to 1 $k_B T$.

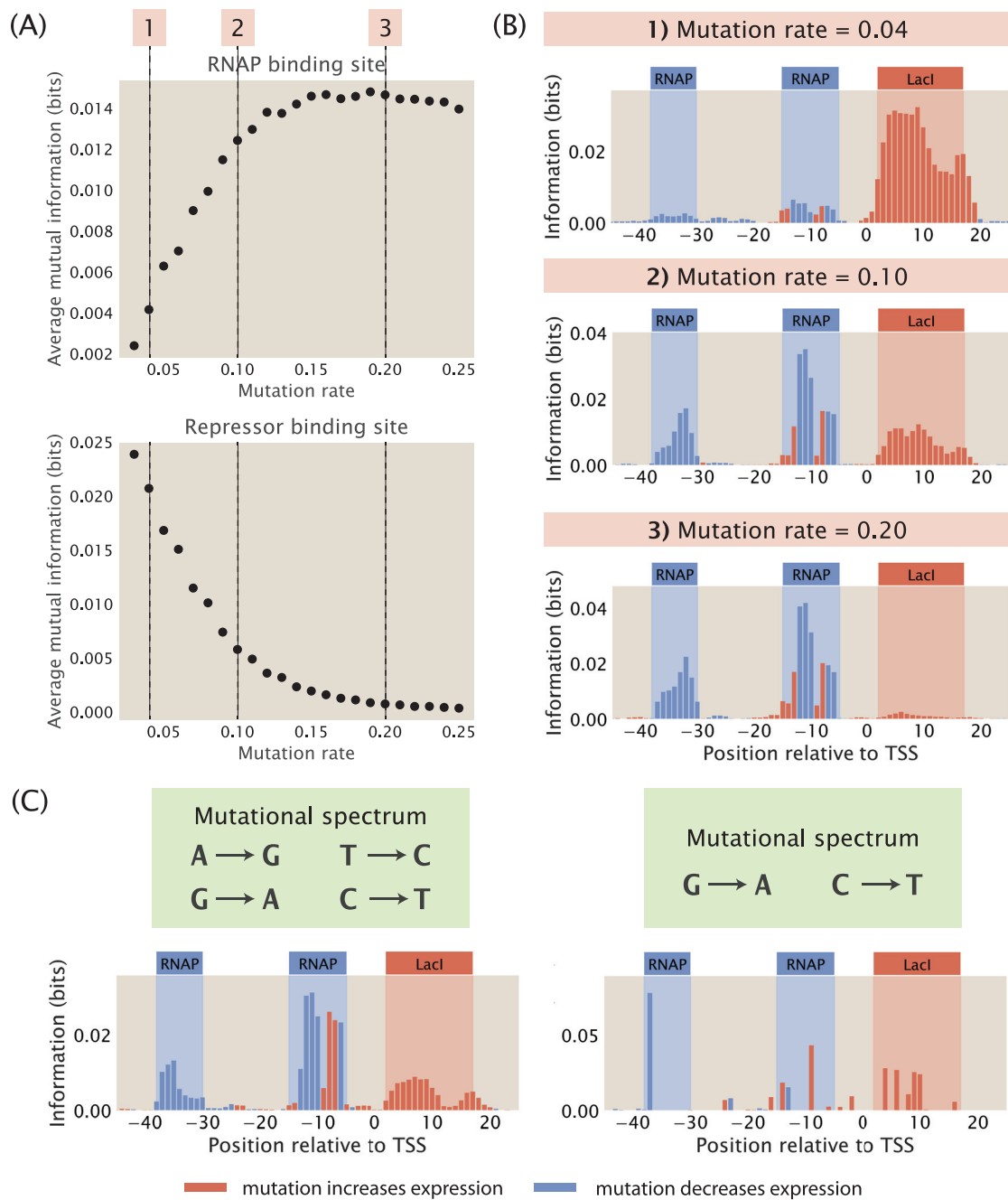

**Fig 5. Changing mutation rate and adding mutational biases.** (A) Changes in the average mutual information at the RNAP and at the repressor binding sites when the mutation rate of the mutant library is increased. Average mutual information is calculated according to Eq 11. Each data point is the mean of average mutual information across 20 synthetic datasets with the corresponding mutation rate. The numbered labels correspond to information footprints shown in (B). (B) Representative information footprints built from synthetic datasets with mutation rates of 0.04, 0.1, and 0.2. (C) Information footprints built from synthetic datasets where the mutant library has a limited mutational spectrum. The left panel shows a footprint where mutations from A to G, G to A, T to C, and C to T are allowed. The right panel shows a footprint where only mutations from G to A and from C to T are allowed.

Boltzmann weights of the repressor and RNAP

$$\kappa = \frac{R \cdot e^{-\beta(\Delta\varepsilon_{\mathrm{rd}} + m_r \Delta\Delta\varepsilon_{\mathrm{rd}})}}{P \cdot e^{-\beta(\Delta\varepsilon_{\mathrm{pd}} + m_p \Delta\Delta\varepsilon_{\mathrm{pd}})}} = \frac{R}{P} \cdot e^{-\beta(\Delta\varepsilon_{\mathrm{rd}} - \Delta\varepsilon_{\mathrm{pd}} + m_r \Delta\Delta\varepsilon_{\mathrm{rd}} - m_p \Delta\Delta\varepsilon_{\mathrm{pd}})}, \tag{12}$$

where $m_r$ and $m_p$ are the number of mutations at the repressor and RNAP binding sites, and $\Delta\Delta\varepsilon_{\mathrm{rd}}$ and $\Delta\Delta\varepsilon_{\mathrm{pd}}$ are the change in binding energies due to each mutation at the repressor and RNAP binding sites. To express $\kappa$ as a function of the mutation rate $\theta$, we can rewrite $m_r$ and $m_p$ as a product of $\theta$ and the lengths of repressor and RNAP binding sites $l_r$ and $l_p$,

$$\kappa = \frac{R}{P} \cdot e^{-\beta E}, \text{ where } E = \Delta\varepsilon_{\mathrm{rd}} - \Delta\varepsilon_{\mathrm{pd}} + \theta(l_r \Delta\Delta\varepsilon_{\mathrm{rd}} - l_p \Delta\Delta\varepsilon_{\mathrm{pd}}) \tag{13}$$

We assume that $\Delta\Delta\varepsilon_{\mathrm{rd}}$ and $\Delta\Delta\varepsilon_{\mathrm{pd}}$ are equal to the average effect of mutations per base pair within each binding site, which can be calculated using the formula

$$\Delta\Delta\varepsilon = \frac{1}{3l} \sum_{i=1}^{l} \sum_{b \neq b_i}^{\Lambda} \varepsilon_{i,b}, \text{ where } \Lambda = \{A, T, C, G\}, \tag{14}$$

where $\varepsilon_{i,b}$ is the energy contribution from position $i$ when the base identity is $b$. As we are using energy matrices where the energies corresponding to the wild-type base identities, $b_i$, are set to 0, we only need to compute the sum of the energy terms for the mutant bases $b \neq b_i \in \Lambda$ at each position. Since there are three possible mutant bases at each site, it follows that to find the average effect of mutations, we divide the sum of the energy matrix by 3 times the length of the binding site $l$.

By applying this formula to the energy matrices in Fig 3B, we see that $\Delta\Delta\varepsilon_{\mathrm{rd}} \approx 2.24 \, k_B T$ and $\Delta\Delta\varepsilon_{\mathrm{pd}} \approx 0.36 \, k_B T$, where $\Delta\Delta\varepsilon_{\mathrm{pd}}$ is averaged over the 20 bases surrounding the -35 and -10 binding sites. Moreover, $l_r = l_p \approx 20$ base pairs. Therefore,

$$l_r \Delta\Delta\varepsilon_{\mathrm{rd}} - l_p \Delta\Delta\varepsilon_{\mathrm{pd}} = 20 \times 2.24 \, k_B T - 20 \times 0.36 \, k_B T = 37.6 \, k_B T. \tag{15}$$

Since the above value is positive, $\kappa$ decreases with increasing mutation rate, making the repressor bound state less likely compared to the RNAP bound state. With the repressor bound state becoming less likely, the signal in the repressor binding site goes down, since mutations changing the binding energy of the repressor change the transcription rate less significantly. As shown in S4 Appendix, when we reduce the effect of mutations on the binding energy of the repressor, we recover the signal at the repressor binding site. Conversely, the average mutual information at the repressor binding site increases when the rate of mutation is decreased. This is because when there are very few mutations, the energy $E$ will be less than 0 and therefore $\kappa$ will be greater than 1. As a result, the repressor will be preferentially bound, which blocks RNAP binding and leads to a low signal at the RNAP binding site. We can recover the signal at the RNAP binding site by increasing the binding energy between RNAP and the wild type promoter, which is also shown in S4 Appendix.

Importantly, the use of $\kappa$ lays the groundwork for finding the optimal rate of mutation in a mutant library. Specifically, we would like to determine the choice of mutation rate that will give us high and balanced signals for both the RNAP and transcription factor binding sites in the information footprints. To find the optimal rate of mutation, we need to satisfy the condition

$$\kappa = \frac{R}{P} e^{-\beta(\Delta\varepsilon_{\mathrm{rd}} - \Delta\varepsilon_{\mathrm{pd}} + \theta(l_r \Delta\Delta\varepsilon_{\mathrm{rd}} - l_p \Delta\Delta\varepsilon_{\mathrm{pd}}))} = 1, \tag{16}$$

which sets repressor and RNAP binding on an equal footing. Plugging in the values of *R*, *P*, and the energy terms and solving for *θ*, we get that $\theta \approx 0.10$. This shows that an intermediate mutation rate is optimal for maintaining high signals at all binding sites. It should be acknowledged that the optimal mutation rate is sensitive to the underlying parameters such as transcription factor copy numbers and binding energies. In S4 Appendix, we numerically compute the optimal mutation rate for copy numbers and binding energies within physiological ranges, and we find that the optimal mutation rate can vary from 0.01 to 0.50. In addition, in S4 Appendix we also calculate the optimal mutation rate for different lengths of the binding sites and a range of $\Delta\Delta\varepsilon_{pd}$ and $\Delta\Delta\varepsilon_{rd}$, and we similarly see that the optimal mutation rate mostly falls within a reasonable range of below 0.5. For an MPRA, our approach provides an estimate for the optimal mutation rate given the parameters of interest. For the remaining analysis shown in this work, we fix the mutation rate at 10%, which is similar to the mutation rate typically used in MPRAs such as Sort-Seq [8] and Reg-Seq [16].

In addition to mutation rate, another important variation in the design of the mutant library is the presence of mutational biases. For example, *in vivo* mutagenesis using the somatic hypermutation enzyme AID from sea lamprey, which is currently the most efficient method of mutagenesis in whole animals, is highly biased towards C to T and G to A mutations [61]. We build mutant libraries that incorporate two different mutational spectrums. In the first case, we allow only swaps between A and G and between C and T. For this library, we observe that the signals at both the RNAP binding site and the repressor binding site are well preserved, as shown in the left panel of Fig 5C. In the second case, we only allow mutations from G to A and from C to T without permitting the reverse mutations, which is the mutational spectrum under the sea lamprey AID system. As shown in the right panel of Fig 5C, due to only two bases being allowed to mutate, only a few, possibly low-effect mutations are observed, making small regions such as the -10 and -35 sites hard to detect. These results inform the minimal mutational spectrum required to produce information footprints and point to the kind of mutagenesis toolkit that should be developed to facilitate MPRAs. For example, as proposed by Falo-Sanjuan et al. [61], the adenine deaminases ABE8e can be used to introduce A to G and T to C mutations to the AID system in order to achieve the footprints shown in the left panel of Fig 5C [62].

**3.1.3 Noise as a function of library size.**   Another parameter that is important for library design is the total number of sequence variants in the mutant library. We build synthetic datasets with varying library sizes and compute the information footprints. To quantify the quality of signal in information footprints, we calculate signal-to-noise ratio, *σ*, according to the formula

$$\sigma = \frac{\langle I \rangle_B}{\langle I \rangle_{NB}}, \text{ where } \langle I \rangle_B = \frac{1}{l_B}\sum_{i \in B} I_i \text{ and } \langle I \rangle_{NB} = \frac{1}{l_{NB}}\sum_{i \in NB} I_i. \tag{17}$$

Here, $I_i$ represents the mutual information at position *i*, *B* is the set of bases within each binding site, *NB* is the set of bases outside the binding sites, $l_B$ is the length of the specific binding site, and $l_{NB}$ is the total length of the non-binding sites. As shown in Fig 6A and 6B, signal-to-noise ratio increases as the library size increases. This may be explained by the "hitch-hiking" effect: since mutations are random, mutations outside of specific binding sites can co-occur with mutations within specific binding sites. As a result, when the library is small, there is an increased likelihood that a mutation outside of specific binding sites and a mutation at a specific binding site become correlated by chance, leading to artificial signal at the non-specific binding sites. The hitch-hiking effects are demonstrated using an analytical example in S5 Appendix.

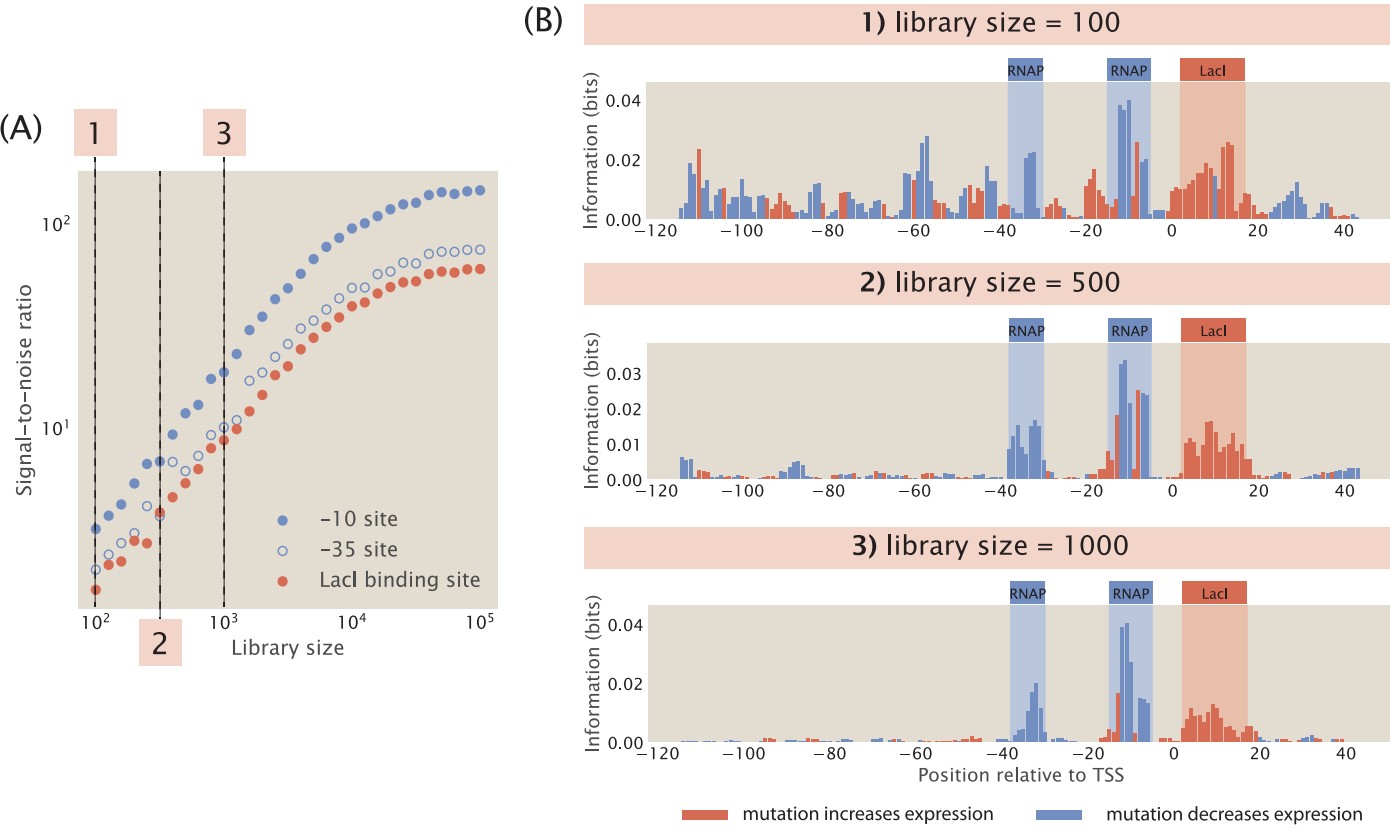

**Fig 6. Noise as a function of library size.** (A) Signal-to-noise ratio increases as library size increases. Signal-to-noise ratio is calculated according to Eq 17. Each data point is the mean of average mutual information across 20 synthetic datasets with the corresponding library size. The numbered labels correspond to footprints in (B). (B) Representative information footprints with a library size of 100, 500, and 1000.

For the remaining analyses shown in this work, we use a library size of 5,000 in order to minimize noise from hitch-hiking effects. In experimental MPRAs, library size needs to be carefully chosen in order to balance high signal-to-noise ratio to the cost of library synthesis. In S5 Appendix, we see that a larger library would make MPRAs cost prohibitive if a large number of promoters are being studied. Our approach provides the means for determining the minimum library size needed to obtain read-outs of reasonable signal-to-noise ratio with the goal of minimizing the cost of these high-throughput experiments.

## 2.2 Perturbing biological parameters in the sequence-specific thermodynamic model

**2.2.1 Tuning the free energy of transcription factor binding.** So far, we have demonstrated that we can build synthetic datasets for the most common regulatory architectures and we have chosen the appropriate mutation rate and library size to construct mutant libraries. Next, we proceed to perturb parameters that affect the probability of RNAP being bound and observe the effects of these perturbations. These analyses will elucidate the conditions required for obtaining clear signals from transcription factor binding events and delineate the limits of MPRA procedures in identifying unannotated transcription factor binding sites.

We again begin by considering the promoter with the simple repression motif, for which the probability of RNAP being bound is given by Eq 1. In *E. coli* grown in minimal media, the

copy number of RNAP is $P \approx 10^3$ [63, 64] and the copy number of the repressor is $R \approx 10$ [65]. The binding energy of RNAP is $\Delta\varepsilon_{\mathrm{pd}} \approx -5\ k_B T$ [52] and the binding energy of the repressor is $\Delta\varepsilon_{\mathrm{rd}} \approx -15\ k_B T$ [66]. Moreover, assuming that the number of non-specific binding sites is approximately equal to the size of the *E. coli* genome, we have that $N_{\mathrm{NS}} \approx 4\times10^6$. Given these values, we can estimate that

$$\frac{P}{N_{\mathrm{NS}}} e^{-\beta\Delta\varepsilon_{\mathrm{pd}}} \approx \frac{10^3}{4 \times 10^6} \times e^5 \approx 0.04 \tag{18}$$

and

$$\frac{R}{N_{\mathrm{NS}}} e^{-\beta\Delta\varepsilon_{\mathrm{rd}}} \approx \frac{10}{4 \times 10^6} \times e^{15} \approx 8. \tag{19}$$

Since $\frac{P}{N_{\mathrm{NS}}} e^{-\beta\Delta\varepsilon_{\mathrm{pd}}} \ll 1$, we can neglect this term from the denominator in Eq 1 and simplify $p_{\mathrm{bound}}$ for the simple repression motif to

$$p_{\mathrm{bound}} = \frac{\dfrac{P}{N_{\mathrm{NS}}} e^{-\beta\Delta\varepsilon_{\mathrm{pd}}}}{1 + \dfrac{R}{N_{\mathrm{NS}}} e^{-\beta\Delta\varepsilon_{\mathrm{rd}}}}. \tag{20}$$

Furthermore, we define the free energy of RNAP binding as

$$F_P = \Delta\varepsilon_{\mathrm{pd}} - k_B T \ln\frac{P}{N_{\mathrm{NS}}} \tag{21}$$

and the free energy of repressor binding as

$$F_R = \Delta\varepsilon_{\mathrm{rd}} - k_B T \ln\frac{R}{N_{\mathrm{NS}}}. \tag{22}$$

Both expressions are written according to the definition of Gibbs free energy, where the first terms correspond to enthalpy and the second terms correspond to entropy. Using these definitions, we can rewrite $p_{\mathrm{bound}}$ as

$$p_{\mathrm{bound}} = \frac{e^{-\beta F_P}}{1 + e^{-\beta F_R}}. \tag{23}$$

In this section, we examine the changes in the information footprints when $F_R$ is tuned. As shown in Fig 7A and 7C, if $F_R$ is increased by reducing the magnitude of $\Delta\varepsilon_{\mathrm{rd}}$ or reducing the copy number of the repressor, we lose the signal at the repressor binding site. For example, compared to the O1 operator, LacI has weak binding energy at the O3 operator, where $\Delta\varepsilon_{\mathrm{rd}} \approx -10\ k_B T$ [66]. Therefore

$$F_R = \Delta\varepsilon_{\mathrm{rd}} - k_B T \ln\frac{R}{N_{\mathrm{NS}}} \approx \left( -10 - \ln\frac{10}{4 \times 10^6} \right) k_B T \approx 3\ k_B T, \tag{24}$$

and

$$e^{-\beta F_R} \approx 0.05. \tag{25}$$

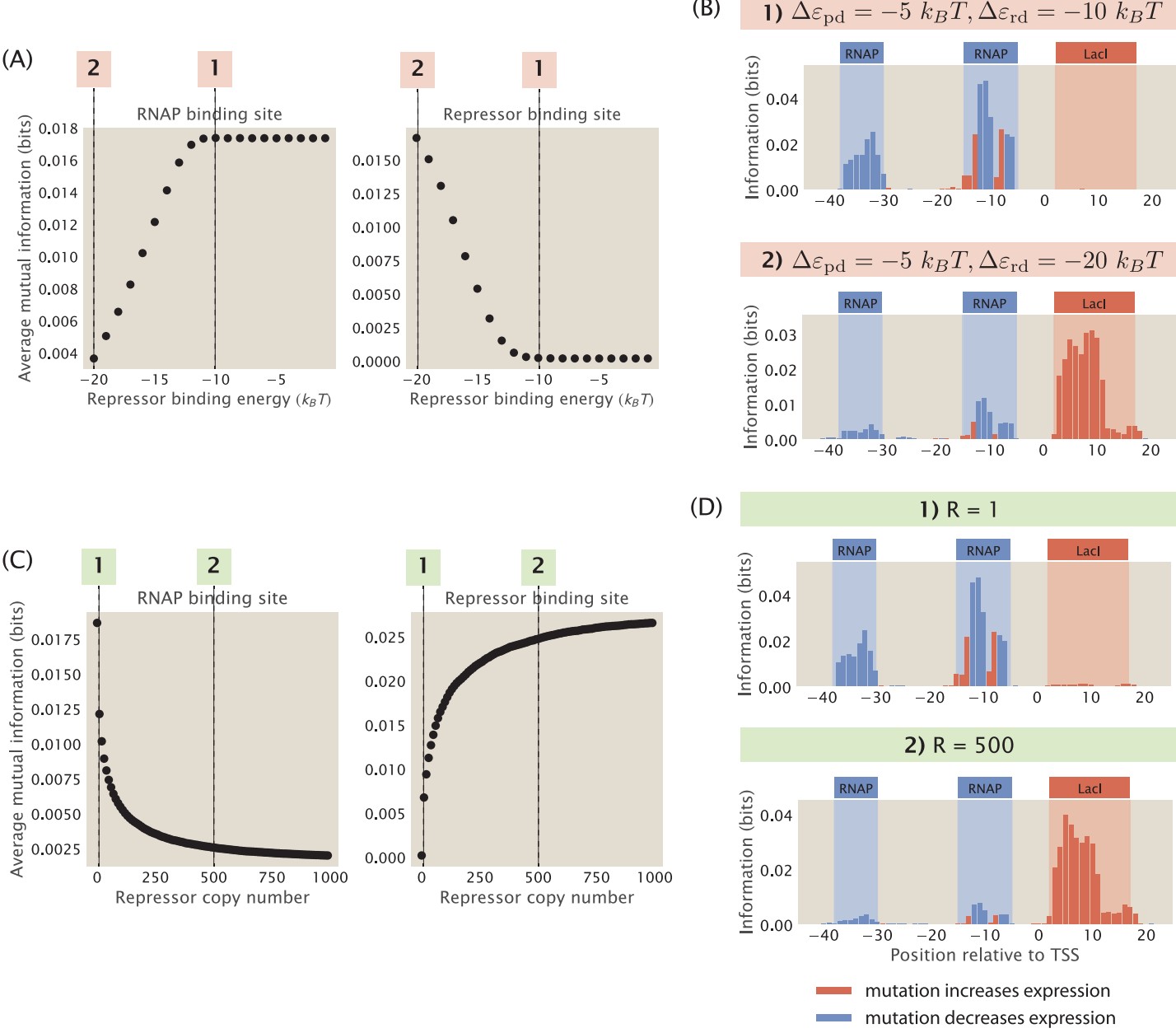

**Fig 7. The strength of the signal at binding sites depends on the free energy of repressor binding.** (A) Increasing the binding energy of the repressor leads to an increase in average mutual information at the RNAP binding site and a decrease in average mutual information at the repressor binding site. $\Delta\varepsilon_{\mathrm{pd}}$ is fixed at -5 $k_BT$, RNAP copy number is fixed at 1000, and repressor copy number is fixed at 10. Each data point is the mean of average mutual information across 20 synthetic datasets with the corresponding repressor binding energy. Numbered labels correspond to footprints in (B). (B) Representative information footprints where $\Delta\varepsilon_{\mathrm{rd}}$ is set to −20 $k_BT$ and −10 $k_BT$. (C) Increasing the copy number of the repressor leads to a decrease in average mutual information at the RNAP binding site and an increase in average mutual information at the repressor binding site. $\Delta\varepsilon_{\mathrm{pd}}$ is fixed at -5 $k_BT$ and $\Delta\varepsilon_{\mathrm{rd}}$ is fixed at -15 $k_BT$. RNAP copy number is fixed at 1000. Each data point is the mean of average mutual information across 20 synthetic datasets with the corresponding repressor copy number. Numbered labels correspond to footprints in (D). (D) Representative information footprints where repressor copy numbers are set to 1 and 500.

In these cases, $e^{-\beta F_R} \ll 1$ and therefore $e^{-\beta F_R}$ can be neglected from the denominator and the probability of RNAP being bound can be simplified to

$$p_{\text{bound}} = e^{-\beta F_P}. \tag{26}$$

This implies that mutations at the repressor binding sites will not have a large effect on $p_{\text{bound}}$ and the mutual information at the repressor binding site will be minimal.

On the other hand, as shown in Fig 7A and 7C, if $F_R$ is decreased either by increasing the magnitude of $\Delta\varepsilon_{\text{rd}}$ or increasing the copy number of the repressor, it leads to a stronger signal at the repressor binding site while significantly reducing the signal at the RNAP binding site. For example, when $\Delta\varepsilon_{\text{rd}} = -20\ k_B T$, we have that

$$F_R = \Delta\varepsilon_{\text{rd}} - k_B T \ln\frac{R}{N_{\text{NS}}} \approx \left(-20 - \ln\frac{10}{4 \times 10^6}\right)\ k_B T \approx -7\ k_B T, \tag{27}$$

and therefore

$$e^{-\beta F_R} \approx 10^3. \tag{28}$$

Here, the Boltzmann weight of the repressor is increased a hundred fold compared to Eq 19. Due to the strong binding of the repressor, mutations at the RNAP binding site do not change expression at measurable levels and therefore the signal is low at the RNAP binding site.

In particular, we see in Fig 7A that when the repressor energy is increased beyond $-11\ k_B T$, the average mutual information at the RNAP binding site saturates and the average mutual information at the repressor binding site remains close to 0. To explain this effect, we again take a look at the ratio between the Boltzmann weights of the repressor and RNAP, the expression for which is stated in Eq 12. Here, we fix the copy number of the repressors and RNAP, the wild-type binding energy of RNAP, the number of mutations, and the effect of mutations. Therefore,

$$\kappa = \frac{R}{P} \cdot e^{-\beta(\Delta\varepsilon_{\text{rd}} - \Delta\varepsilon_{\text{pd}} + m_R \Delta\Delta\varepsilon_{\text{rd}} - m_P \Delta\Delta\varepsilon_{\text{pd}})} \tag{29}$$

$$= \frac{10}{1000} \cdot e^{-\beta\Delta\varepsilon_{\text{rd}} - 5 - 2\times2.24 + 2\times0.36} \tag{30}$$

$$= 1.5 \times 10^{-6} \cdot e^{-\beta\Delta\varepsilon_{\text{rd}}}. \tag{31}$$

We assume that $\kappa$ needs to be at least 0.1 for there to be an observable signal at the repressor binding site. Solving for $\varepsilon_{\text{rd}}$ using the above equation, we have that $\Delta\varepsilon_{\text{rd}} \approx -11\ k_B T$. This matches with our observation that the signal stabilizes when $\Delta\varepsilon_{\text{rd}} > -11\ k_B T$. Taken together, these result invite us to rethink our interpretation of MPRA data, as the lack of signal may not necessarily indicate the absence of a binding site, but it may instead be a result of weak binding or low transcription factor copy number.

**2.2.2 Changing the regulatory logic of the promoter.** In the previous section, we examined the changes in information footprints when we tune the copy number of the repressors under the simple repression regulatory architecture. The effect of transcription factor copy numbers on the information footprints is more complex when a promoter is regulated by multiple transcription factors. In particular, the changes in the information footprints depend on the regulatory logic of the promoter. To see this, we consider a promoter that is regulated by two repressors. For a double-repression promoter, there are many possible regulatory logics; two of the most common ones are AND logic and OR logic. As shown in S2 Appendix, if the

two repressors operate under AND logic, both repressors are required to be bound for repression to occur. This may happen if each of the two repressors bind weakly at their respective binding sites but bind cooperatively with each other. On the other hand, if the two repressors operate under OR logic, then only one of the repressors is needed for repression.

We generate synthetic datasets for an AND-logic and an OR-logic double-repression promoter that are regulated by two repressors with copy numbers $R_1$ and $R_2$. As shown in Fig 8A and 8B, under AND logic, there is no signal at either of the repressor binding sites when $R_1 = 0$. This matches our expectation because AND logic dictates that a single repressor is not able to reduce the level of transcription by itself. We recover the signal at both of the repressor binding sites even when $R_1 = 1$. Interestingly, when $R_1 > R_2$, the signal at the first repressor binding site is lower than at the second repressor binding site. This may be because the higher $R_2$ compensates for the effects of mutations and therefore expression levels are affected to a greater extent by mutations at the first repressor binding site than by mutations at the second repressor binding site. In comparison, since the two repressors act independently under OR logic, the signal at the second repressor binding site is preserved even when $R_1 = 0$. Moreover, the state where the first repressor represses transcription competes with the state where the second repressor represses transcription. As a result, when $R_1$ is increased, the signal at the

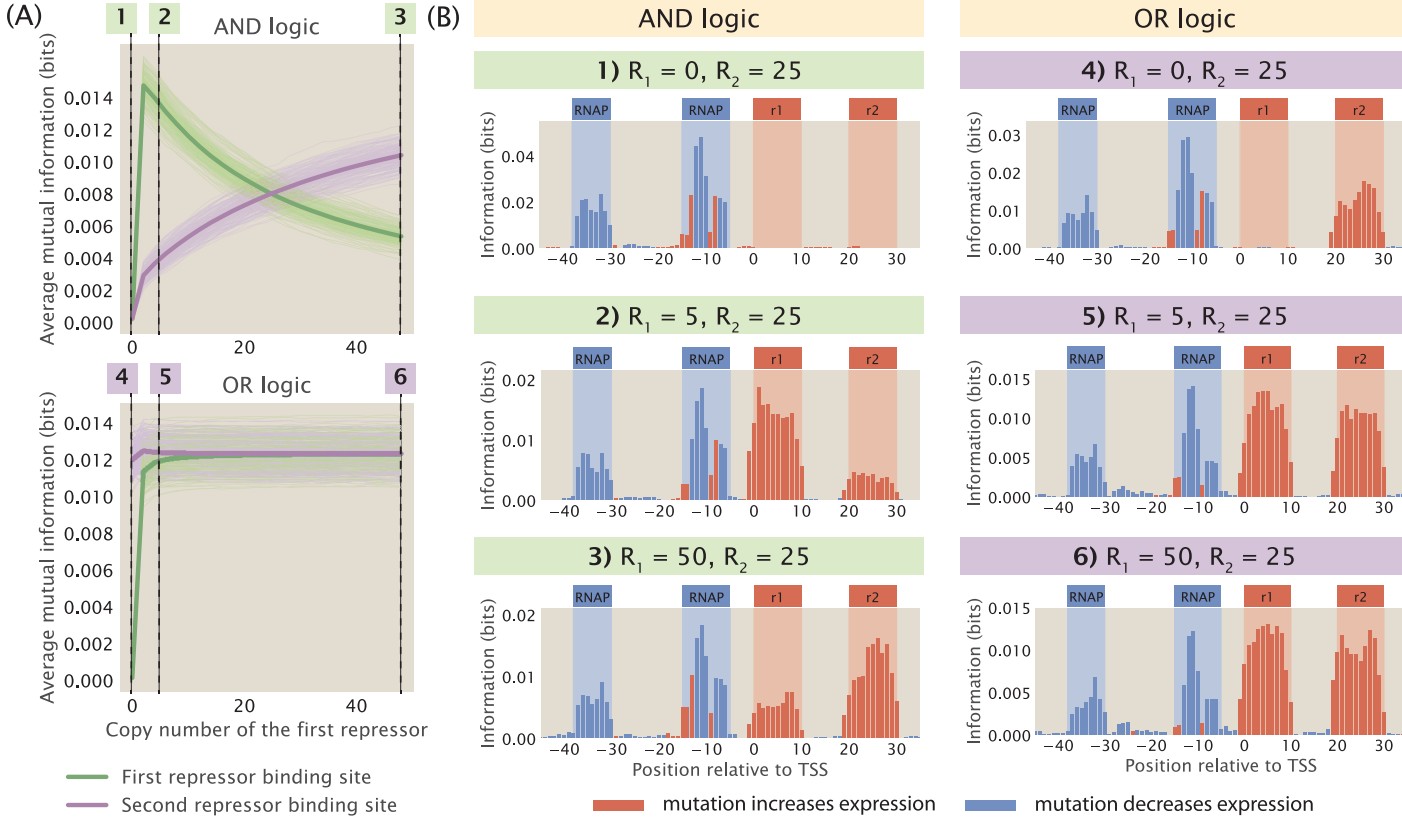

**Fig 8. Changing repressor copy number for a double-repression promoter.** (A) Changing the copy number of the first repressor under AND logic and OR logic affects the signal at both repressor binding sites. For the energy matrices of the repressors, the interaction energy between the repressor and a site is set to 0 $k_B T$ if the site has the wild-type base identity and set to 1 $k_B T$ if the site has the mutant base identity. The interaction energy between the repressors is set to $-5$ $k_B T$. 200 synthetic datasets are simulated for each copy number. We observe that the average mutual information at binding sites has high variability across synthetic datasets, especially under OR logic. To show variability, the trajectory for each of the synthetic dataset is shown as an individual light green or light purple curve. The average trajectories across all 200 synthetic datasets are shown as the bolded green curves and the bolded purple curves. The numbered labels correspond to footprints in (B). (B) Representative information footprints of a double repression promoter under AND and OR logic.

first repressor binding site is increased whereas the signal at the second repressor binding site decreases.

As illustrated in S6 Appendix, the procedure described above can be extended to other regulatory logics such as the XOR gate and other promoter architectures such as a promoter that is regulated by two activators. These results are informative in the context of transcription factor deletion, which is a key approach for identifying and verifying which transcription factor binds to the putative binding sites discovered in MPRAs [16]. The final copy number of the transcription factor depends on which experimental method is chosen to perform the deletion. If the gene coding for the transcription factor is knocked out, no transcription factor will be expressed and the transcription factor copy number will be 0. Therefore, by comparing the footprints from the wild-type strain and the transcription factor deletion strain, we can locate the site where the signal disappears and deduce which transcription factor is bound at that site. On the other hand, if knock-down methods such as RNA interference are used, some leaky expression may take place and the transcription factor copy number may be low but non-zero. In this case, while one may expect that the signal at the corresponding binding sites would also disappear, our simulations suggest that there may not be appreciable differences in the footprints from the wild-type strain and the deletion strain. This would be an important point of consideration in the interpretation of MPRA read-outs when knock-down methods are used to map transcription factors to binding sites.

**2.2.3 Competition between transcription factor binding sites.** Thus far, we have assumed that each transcription factor only has one specific binding site in the genome. However, many transcription factors bind to multiple promoters to regulate the transcription of different genes. For example, cyclic-AMP receptor protein (CRP), one of the most important activators in *E. coli*, regulates 330 transcription units [67]. Therefore, it is important to understand how the relationship between sequence and binding energy changes when the copy number of the transcription factor binding site is changed.

Binding site copy number is also highly relevant in the context of experimental MPRA protocols. When *E. coli* is the organism of interest, there are two main ways of delivering mutant sequences into the cell. The first is by directly placing the mutant promoter into the genome using genome integration methods such as ORBIT [68, 69]. In this case, we would have one copy of the mutant promoter in the cell. The second method is to transform the bacterial cells with plasmids carrying the promoter variant. If this approach is used, the number of mutant binding sites will be equal to the copy number of the plasmids. We would like to understand precisely how the signal in the resulting information footprint differs between a genome integrated system and a plasmid system.

To build a synthetic dataset that involves more than one transcription factor binding site, we once again begin by building a thermodynamic model to describe the different binding events. However, in the canonical thermodynamic model that we utilized earlier, introducing multiple transcription factor binding sites would lead to a combinatorial explosion in the number of possible states. To circumvent this issue, we introduce an alternative approach to build thermodynamic models based on the concept of chemical potential, which is explained in detail in S7 Appendix.

Using the method of chemical potential, we construct synthetic datasets with different repressor binding site copy numbers. As shown in Fig 9A, as the copy number of repressor binding sites is increased, the signal at the repressor binding site decreases rapidly and eventually stabilizes at a near-zero value. In particular, as shown in Fig 9B, in a genome integrated system where there is only one copy of the repressor binding site, there is clear signal. On the other hand, in a plasmid system where the copy number of the binding site is greater than the copy number of the repressor, the signal for the repressor disappears. Intuitively, this can be

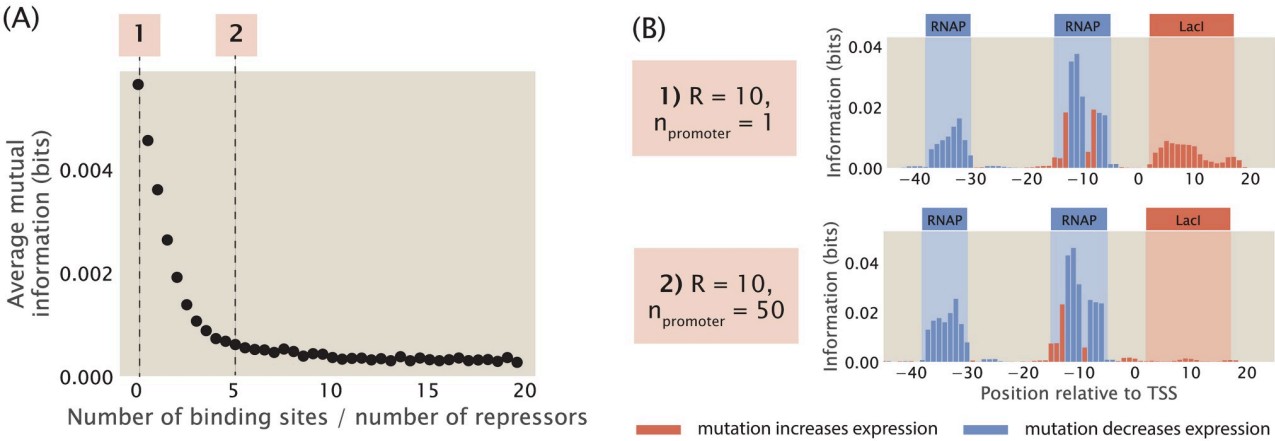

**Fig 9. Changing the copy number of transcription factor binding sites.** (A) Average mutual information at the repressor binding site decreases when the number of repressor binding sites is increased. Repressor copy number is set to 10 for all data points. Each data point is the mean of average mutual information across 20 synthetic datasets with the corresponding number of repressor binding sites. Numbered labels correspond to footprints in (B). (B) Representative information footprints for cases where there is only 1 repressor binding site and when there are 50 repressor binding sites.

explained by the so-called titration effects [70], where the additional binding sites titrate away repressors, which reduces the effective number of repressors in the system. As a result, the expression of the reporter gene no longer reflects transcriptional regulation by the repressor. This reduces the effect of mutations at the repressor binding site on expression levels, which leads to low mutual information between mutations and base identities at the repressor binding site.

In wild type *E. coli*, the median ratio of transcription factor copy number and binding site copy number is around 10 [71], and therefore titration effects are unlikely to diminish the signals in information footprints when the sequence variant is integrated into the genome. On the other hand, if a plasmid system is used, it is advisable to make use of a low copy number plasmid. Although we have no knowledge of which transcription factor is potentially regulating the gene of interest and therefore we do not know a priori the copy number of the transcription factor, using a low copy number plasmid has a greater chance of ensuring that the copy number of the transcription factor binding sites is no greater than the copy number of the putative transcription factor.

**2.2.4 Changing the concentration of the inducer.** So far, in the regulatory architectures that involve repressor binding, we have only considered repressors in the active state, whereas in reality the activity of the repressors can be regulated through inducer binding. Specifically, according to the Monod–Wyman–Changeux (MWC) model, the active and inactive states of a repressor exist in thermal equilibrium and inducer binding may shift the equilibrium in either direction [32]. If inducer binding shifts the allosteric equilibrium of the repressor from the active state towards the inactive state, the repressor will bind more weakly to the promoter. This will increase the probability of RNAP being bound and therefore lead to higher expression. In other words, increasing inducer concentration has similar effects to knocking out the repressor from the genome. For example, when lactose is present in the absence of glucose, lactose is converted to allolactose, which acts as an inducer for the Lac repressor and increases the expression of genes in the lacZYA operon. Conversely, some inducer binding events may also shift the equilibrium of a repressor from the inactive state towards the active state. One example is the Trp repressor, which is activated upon tryptophan binding and represses gene expression. Here, we use the example of the lacZYA operon and demonstrate how signals in

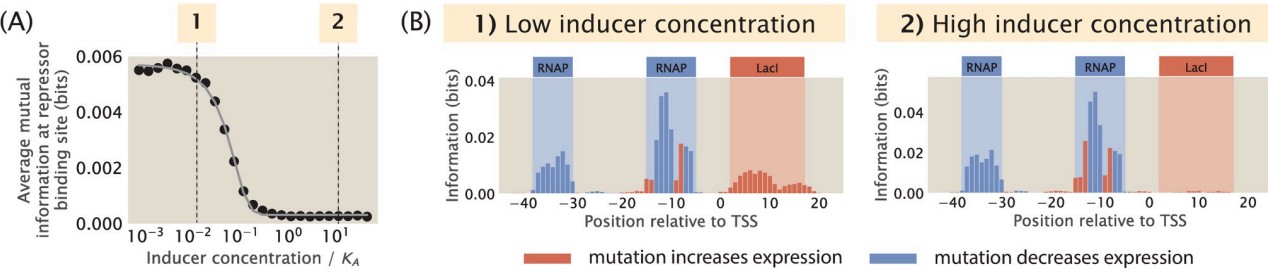

**Fig 10. Changing the concentration of the inducer.** (A) Average mutual information at the repressor binding site decreases as the inducer concentration increases. $p_{bound}$ for the promoter with an inducible repressor is derived in S8 Appendix. Here, we use the following thermodynamic parameters: the dissociation constant of the inducer at the binding pockets of the active repressor $K_A = 139 \times 10^{-6}$ M, the dissociation constant of the inducer at the binding pockets of the inactive repressor $K_I = 0.53 \times 10^{-6}$ M, and the structural energy difference between the active repressor and the inactive repressor $\Delta \varepsilon_{AI} = 4.5\ k_B T$. The thermodynamic parameters were inferred by Razo-Mejia et al. from predicted IPTG induction curves [72]. The inducer concentration on the x-axis is normalized with respect to the value of $K_A$. Each data point is the mean of average mutual information across 20 synthetic datasets with the corresponding inducer concentration. The numbered labels correspond to footprints in (B). (B) Representative information footprints with low inducer concentration ($10^{-6}$ M) and high inducer concentration ($10^{-3}$ M).

the information footprint depend on the concentration of the allolactose inducer in the system.

In S8 Appendix, we describe our approach to calculate the probability of RNAP being bound when there is an inducible transcription factor in the system. Using this approach, we built synthetic datasets for a promoter with the simple repression regulatory architecture with an inducible repressor and a range of inducer concentrations. The inducer concentration is normalized with respect to $K_A$, the dissociation constant of the inducer binding at the binding pockets of the active repressor. As shown in Fig 10A and 10B, when the concentration of the inducer is increased from $10^{-2}\ K_A$ to $1\ K_A$, the average signal at the repressor binding site decreases. Interestingly, the average signal is not reduced further when the concentration is increased beyond the value of $K_A$. As shown in S8 Appendix, similar results are observed in the case of a simple activation promoter with an inducible activator.

These results show that the concentration of inducers can determine whether we will obtain a signal at the transcription factor binding site. This underscores the importance of performing experimental MPRAs under different growth conditions using appropriate inducer concentrations to ensure that we can identify binding sites that are bound by transcription factors induced by specific cellular conditions. These efforts may fill in the gap in knowledge on the role of allostery in transcription, which so far has been lacking attention from studies in the field of gene regulation [73].

## 2.3 Identifying transcription factor binding sites from information footprints

### 2.3.1 Noise due to stochastic fluctuations of RNAP and transcription factor copy number.
When examining summary statistics of MPRAs such as information footprints, one way to annotate transcription factor binding sites is to identify regions where the signal is significantly higher than background noise. Therefore, to be able to precisely and confidently identify RNAP and transcription factor binding sites from information footprints, the footprint is required to have a sufficiently high signal-to-noise ratio. However, this may not always be the case. For example, the footprint shown in Fig 11 was obtained for the *mar* operon by Ireland et al. [16]; while the signal at the RNAP binding sites and the -20 MarR binding sites are clearly higher than the background noise, the signals at the Fis, MarA, and +10 MarR binding sites may easily be mistaken for noise.

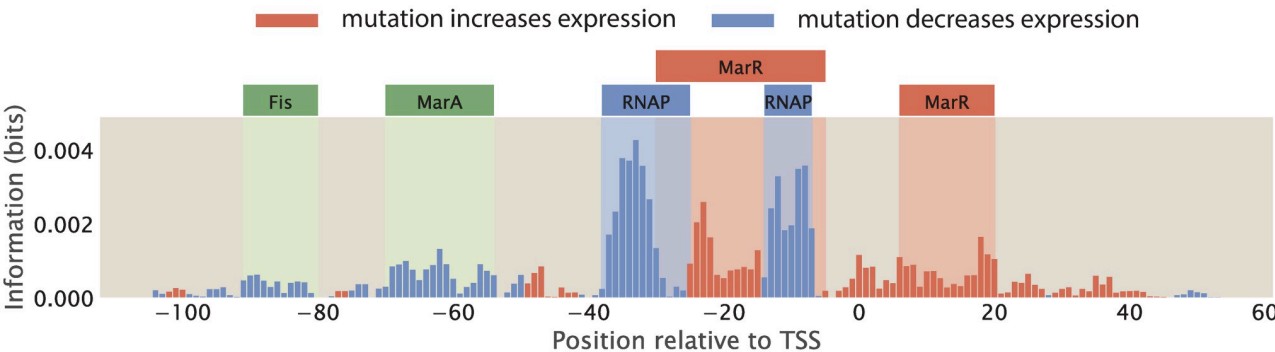

**Fig 11. Annotating transcription factor binding sites by identifying sites with high signal.** The footprint of the *mar* operon, produced by Ireland et al [16]. The binding sites are annotated based on known RNAP and transcription factor binding sites; the signal at some of the binding sites, such as the Fis and MarA binding sites, are not distinguishable from background noise.

In Sec 2.1.3, we examined how the size of the mutant library may affect the level of noise in information footprints. Here, we continue to examine other factors that may affect signal-to-noise ratio. We first consider possible sources of experimental noise, including PCR amplification bias and random sampling effects during RNA-Seq library preparation and RNA-Seq itself. In S9 Appendix, we simulate synthetic datasets with various levels of experimental noise and we show that these experimental processes do not lead to significant levels of noise outside of the specific binding sites. Another potential source of noise is from the biological noise that contributes to stochastic fluctuations in expression levels. These sources of biological noise can be broadly categorized into intrinsic noise and extrinsic noise [74–76]. Intrinsic noise arises from the inherent stochasticity in the process of transcription, such as changes in the rate of production or degradation of mRNA. On the other hand, extrinsic noise arises from cell-to-cell variation in the copy number of transcriptional machineries such as RNAP and transcription factors. It has been shown that extrinsic noise occurs on a longer timescale and has a greater effect on phenotypes than intrinsic noise [76]. Here, we investigate whether extrinsic noise has an effect on information footprints.

We build synthetic datasets for a promoter with the simple repression architecture using the same procedure as before, except we no longer specify the copy number of RNAPs and repressors as a constant integer. Instead, as described in S10 Appendix, we randomly draw the copy numbers of RNAPs and repressors from a Log-Normal distribution, which is the distribution that the abundance of biomolecules often follows [77]. Brewster et al. used the dilution method to measure the stochastic fluctuations in transcription factor copy numbers due to asymmetrical partitioning during cell division and found that transcription factor copy numbers typically vary by less than the 20% of the mean copy numbers [70]. Moreover, proteomic measurements suggest that the coefficient of variation for copy numbers is less than 2 across diverse growth conditions [71, 78]. Here, we build Log-Normal distributions with a range of coefficients of variation, which cover both the reported levels of extrinsic noise as well as coefficients of variation as high as 100, which is physiologically extreme. As shown in Fig 12A and 12B, we observe that signal-to-noise ratio decreases when extrinsic noise is higher. However, we can still distinguish between signal and noise even when we specify a large coefficient of variation. Moreover, as seen in S10 Appendix, even when the signal at the repressor binding site is low due to other factors such as low repressor binding energy, the signal from both binding sites is still distinguishable from the noise caused by copy number fluctuations. In addition, as shown in S10 Appendix, signal-to-noise ratio remains high in different regulatory

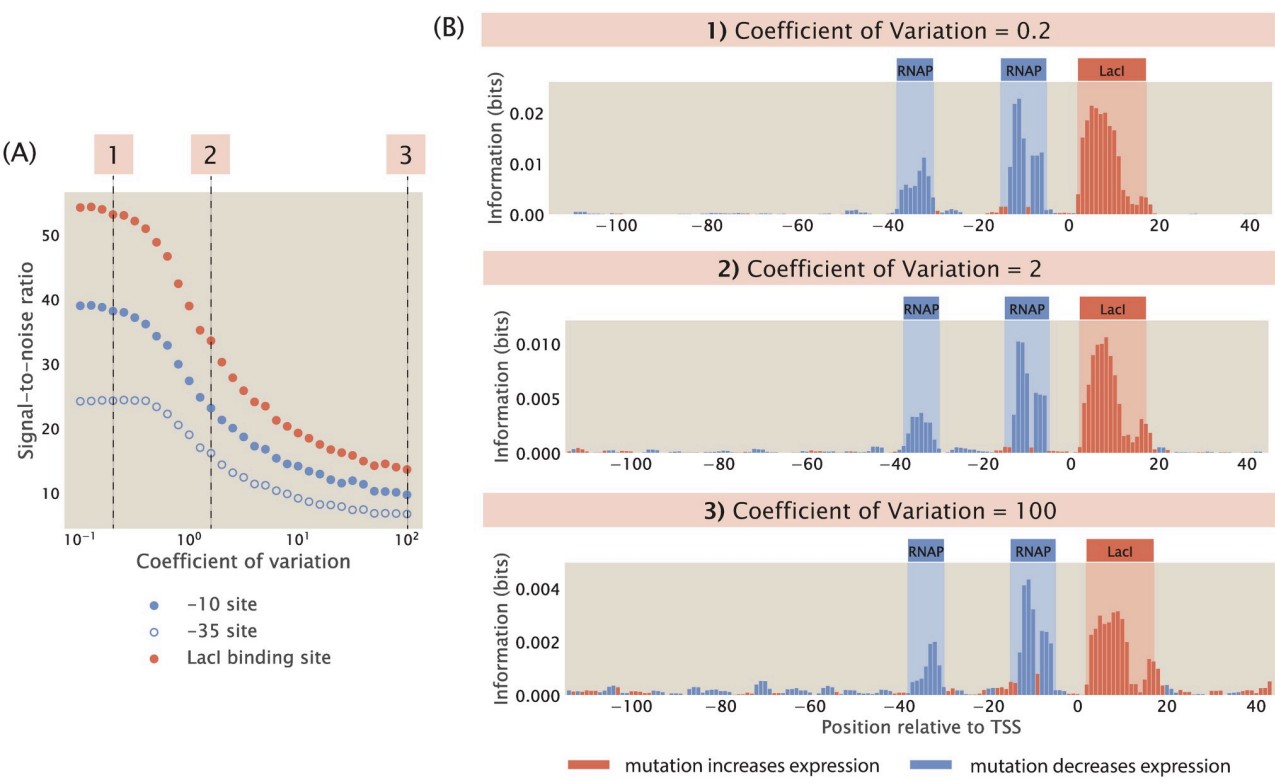

**Fig 12. Adding extrinsic noise to synthetic datasets.** (A) Increasing extrinsic noise lowers the signal-to-noise ratio in information footprints. For all synthetic datasets, the copy numbers of RNAP and repressors are drawn using the Log-Normal distributions described in S10 Appendix. In the Log-Normal distributions, $\mu$ is set to 5000 for RNAPs and 100 for repressors. Each data point is the mean of average mutual information across 100 synthetic datasets with the corresponding coefficient of variation. The numbered labels correspond to footprints in (B). (B) Representative information footprints with three levels of extrinsic noise.

architectures in the presence of extrinsic noise. These results suggest that information footprints as an MPRA read-out are robust to extrinsic noise.

This phenomenon may be explained by the fact that the changes in binding energies due to mutations in the promoter sequence have a much greater contribution to the probability of RNAP being bound than do changes in the copy number of transcription factors. Assuming that RNAP binds weakly to the promoter, the expression for $p_{\text{bound}}$ in Eq 1 can be simplified to

$$
p_{\text{bound}} = \frac{\frac{P}{N_{\text{NS}}} e^{-\beta \Delta \varepsilon_{\text{pd}}}}{1 + \frac{R}{N_{\text{NS}}} e^{-\beta \Delta \varepsilon_{\text{rd}}}} .
\tag{32}
$$

Based on the experimentally measured energy matrix for LacI [52], the average increase in $\Delta \varepsilon_{\text{rd}}$ due to one mutation is approximately 2 $k_B T$. The LacI binding site is around 20 base pairs long. Therefore, with a 10% mutation rate, there are on average 2 mutations within the LacI binding site, and the total change in $\Delta \varepsilon_{\text{rd}}$ would be approximately 4 $k_B T$. This leads to a $e^{-\beta \Delta \Delta \varepsilon_{\text{rd}}} = 0.01$ fold change in the magnitude of $\frac{R}{N_{\text{NS}}} e^{-\beta \Delta \varepsilon_{\text{rd}}}$. This means that the copy number of the repressor would have to change by a factor of 100 to overcome the effect of mutations, and this is not possible through fluctuations due to extrinsic noise. Therefore, extrinsic noise by

itself will not lead to a sufficiently large change in expression levels to affect the signals in information footprints.

**2.3.2 Non-specific binding along the promoter.**   In the earlier sections of this paper, our thermodynamic models only allow RNAP to bind at one position along the promoter. However, in reality, non-specific binding events along the rest of the promoter also occur, albeit at low frequencies. To investigate the effect of non-specific binding on information footprints, we build a thermodynamic model that allow for RNAP binding at every possible position along the promoter as a unique state. The states-and-weights diagram of this expanded thermodynamic model is illustrated in Fig 13A. The weight of each state is calculated by mapping the energy matrix to the corresponding non-specific binding site sequence at each position along the promoter. As shown in Fig 13B top panel, in general, non-specific binding only leads to a small amount of noise. Similar to the case of extrinsic noise, this source of noise is not at a sufficiently high level to interfere with our ability to delineate binding site positions. This implies that reducing the hitch-hiking effects described in Sec 2.1.3 should be the primary focus when a high signal-to-noise ratio is desired.

On the other hand, a more interesting phenomenon when RNAP is allowed to bind along the entire promoter is the presence of strong signal at non-canonical binding sites. In particular, signal may arise at these sites due to the presence of TATA-like motifs. RNAPs with $\sigma_{70}$ typically bind to the TATA box, which is a motif with the consensus sequence TATAAT located at the -10 site along the promoter. However, since the sequence motif is short, it is likely that TATA-like motifs with a short mutational distance from the TATA-box sequence may exist away from the -10 site. A mutation may easily convert these motifs into a functional TATA-box, allowing RNAP to initiate transcription from a different transcription start site. In the bottom information footprint of Fig 13B, the promoter is engineered to contain a TATA-like motif upstream of the canonical binding sites. As shown in the footprint, this leads to a strong signal at the -100 and -75 positions. This analysis unveils a feature of information

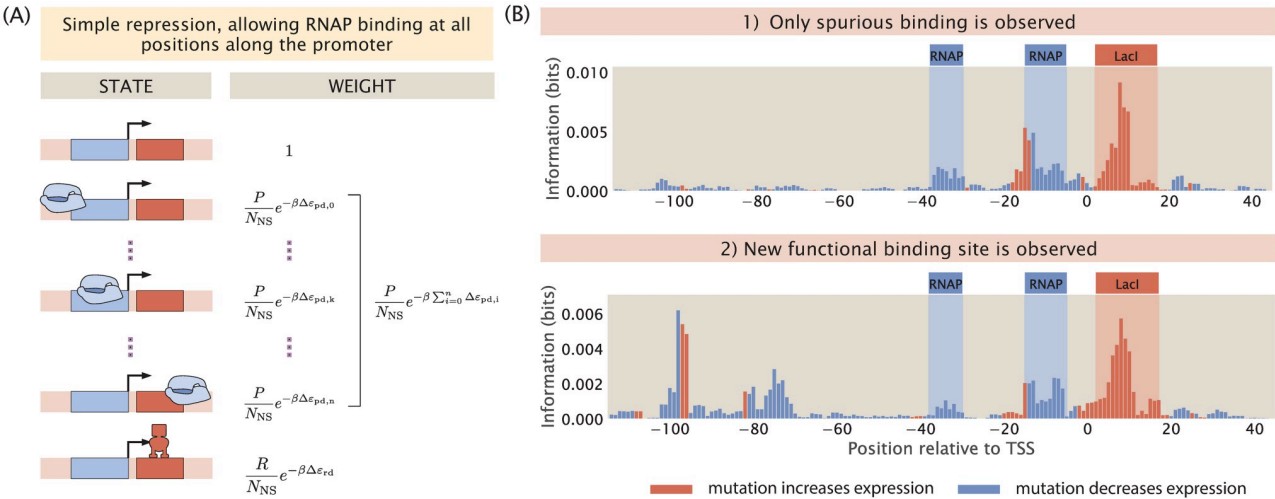

**Fig 13. Non-specific RNAP binding can create low levels of noise and lead to non-canonical functional binding sites.** (A) States-and-weights diagram of a simple repression promoter where spurious RNAP binding is allowed. For each of the RNAP spurious binding events, the binding energy, $\Delta\varepsilon_{\text{pd},i}$, is computed by mapping the RNAP energy matrix to the spurious binding site sequence. The index $i$ corresponds to the position of the first base pair to which RNAP binds along the promoter. 0 is at the start of the promoter sequence; $k$ is at the canonical RNAP binding site; $n = 160 - l_p$ is the index of the most downstream binding site where the promoter is assumed to be 160 base pairs long and $l_p$ is the length of the RNAP binding site. (B) Information footprints of a promoter under the simple repression regulatory architecture with non-specific binding (top) and with a new functional binding site (bottom). The bottom plot is created by inserting the sequence "TAGAAT", which is one letter away from the TATA-box sequence, at the -80 position.

footprints that should be considered in the interpretation of signal and noise in real-world MPRA datasets.

**2.3.3 Overlapping binding sites.**   Other than a low signal-to-noise ratio, another factor that may contribute to the challenge of deciphering regulatory architectures is the presence of overlapping binding events. This is especially common with RNAP and repressor binding sites, since a common mechanism by which repressors act to reduce expression is by binding to the RNAP binding site and thereby sterically blocking RNAP binding. For example, in Fig 11, we can see that the MarR binding site overlaps with the RNAP binding site. Assuming perfect binding sites, a mutation in the RNAP binding site will decrease expression and a mutation in the repressor binding site will increase expression. However, if the binding sites overlap, we expect either the signals from the two binding events will cancel out or one signal will dominate the other. Here, we build synthetic datasets with different degrees of overlap between RNAP and repressor binding sites, and we examine how much overlap can be tolerated before the two binding sites are no longer distinguishable from each other.

In Fig 14, we show a series of information footprints and expression shift matrices where the repressor is slid along the full range of the RNAP binding site. The promoter sequences are engineered to maximize binding strengths based on the energy matrices of the RNAP and the repressor shown in Fig 3B. At positions along the promoter that are only bound by RNAP, the base that minimizes the binding energy of the RNAP is chosen. The same applies for positions that are only bound by the repressor. On the other hand, at overlapping binding sites, base identities that minimize the total binding energy by the RNAP and the repressor are chosen. The base identities of the rest of the promoter sequence are selected at random. As shown in Fig 14A, the signals from the two binding events are clearly segmented and distinguishable from each other when less than 50% of the binding sites are overlapping. In contrast, when the vast majority of the base positions are overlapping, signal from repressor binding dominates the signal from RNAP binding. In the expression shift matrices shown in Fig 14B, when the repressor binds directly on top of the -10 RNAP binding site, some signal from RNAP binding is still preserved. However, these signals are not strong enough to highlight the presence of an RNAP binding site. Without prior knowledge that RNAP binds at this position, such a footprint could lead to the erroneous conclusion that only repressors bind to this site. These analyses demonstrate the challenge of deciphering regulatory architectures in the presence of overlapping binding sites. This may be overcome by tuning growth conditions to reduce binding by some of the overlapping binding partners, such that we can obtain cleaner footprints with signal indicating individual binding events.

## 2.4 Building synthetic datasets under non-equilibrium conditions

So far in this paper, one important assumption underlying our thermodynamic models is that the processes involved in transcription initiation are in quasi-equilibrium. The success of thermodynamic models in predicting experimental outcomes in previous works lends credibility to the use of equilibrium models in prokaryotic systems such as *E. coli* [52, 59, 66, 70, 72]. However, in transcriptional regulation more generally, there are known to be energy consuming processes such as phosphorylation and nucleosome remodelling. Therefore, it is important to consider how our computational pipeline may be extended to systems where detailed balance is broken.

To account for non-equilibrium processes in the construction of synthetic MPRA datasets, we invoke the graph-theoretic approach proposed by Gunawardena [79] and used by Mahdavi, Salmon et al. [80] to calculate the probability of transcriptionally active states, which allows us to consider the full kinetic picture of trancription initiation. In S11 Appendix, we detail our

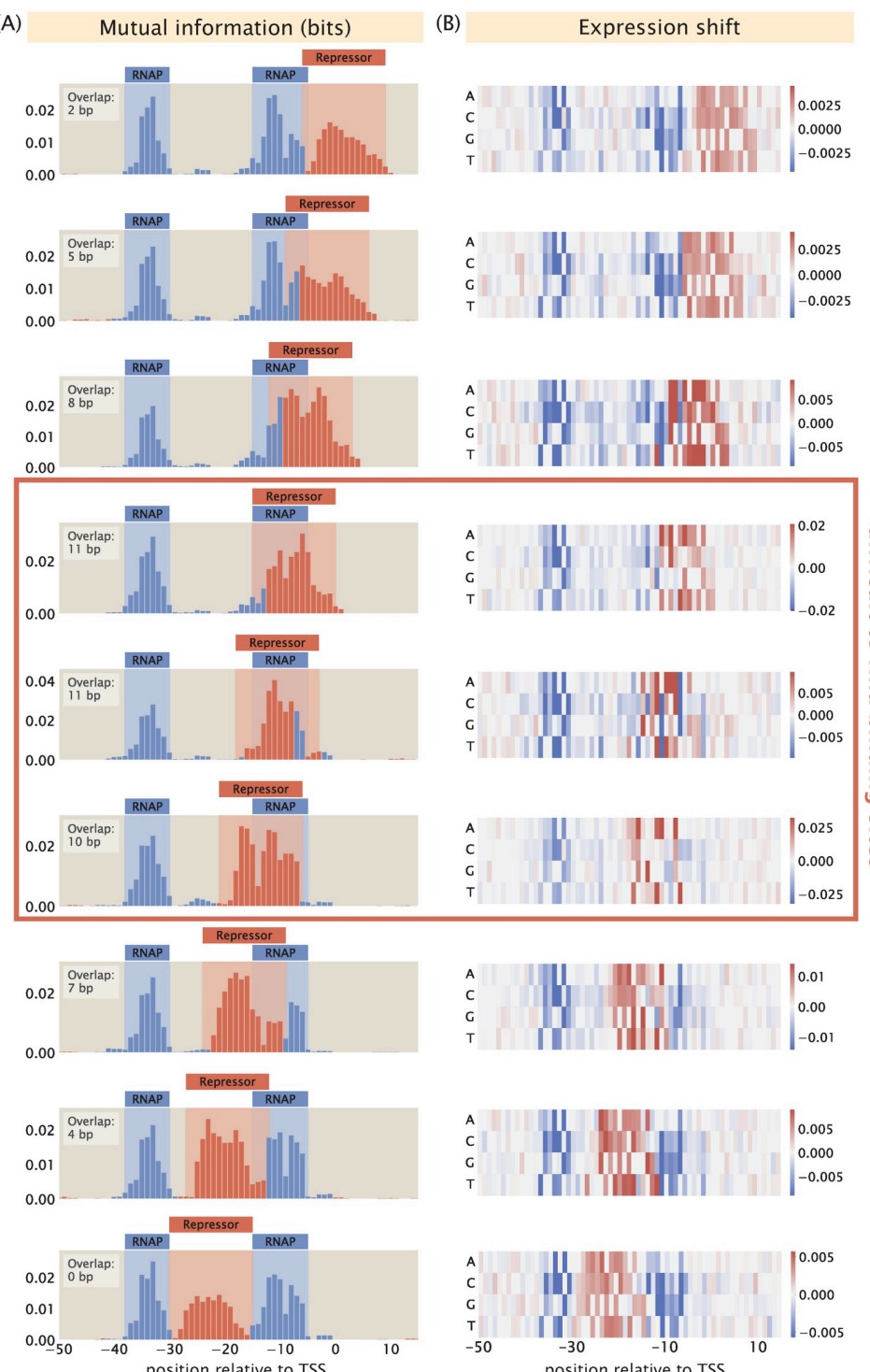

**Fig 14. Changing the degree of overlap between the RNAP and repressor binding sites.** (A—B) Information footprints and expression shift matrices of a simple repression promoter with overlapping binding sites. The promoters are designed to maximize binding strength given the known energy matrices of the RNAP [52] and LacI [53]. The degree of overlap in the information footprints and expression shift matrices in each row is noted at the upper left hand corner of the footprints.

approach for calculating the probability of RNAP being bound under non-equilibrium using the Matrix Tree Theorem. Using this approach, we can predict the expression levels of the promoter variants without imposing equilibrium constraints. This makes it possible for us to build a synthetic dataset that does not rely on the quasi-equilibrium assumptions, and will provide further clues about how to interpret MPRA datasets. As shown in Fig 15B, the information footprint built from the graph-theoretic synthetic dataset under equilibrium is comparable to information footprints built from our thermodynamic models.

As shown in Fig 15A, when we break detailed balance at the edge where RNAP unbinds from the state where both the activator and RNAP are bound, varying $U$ and the concentration of the activator leads to interesting behaviour in the information footprints. As shown in panel

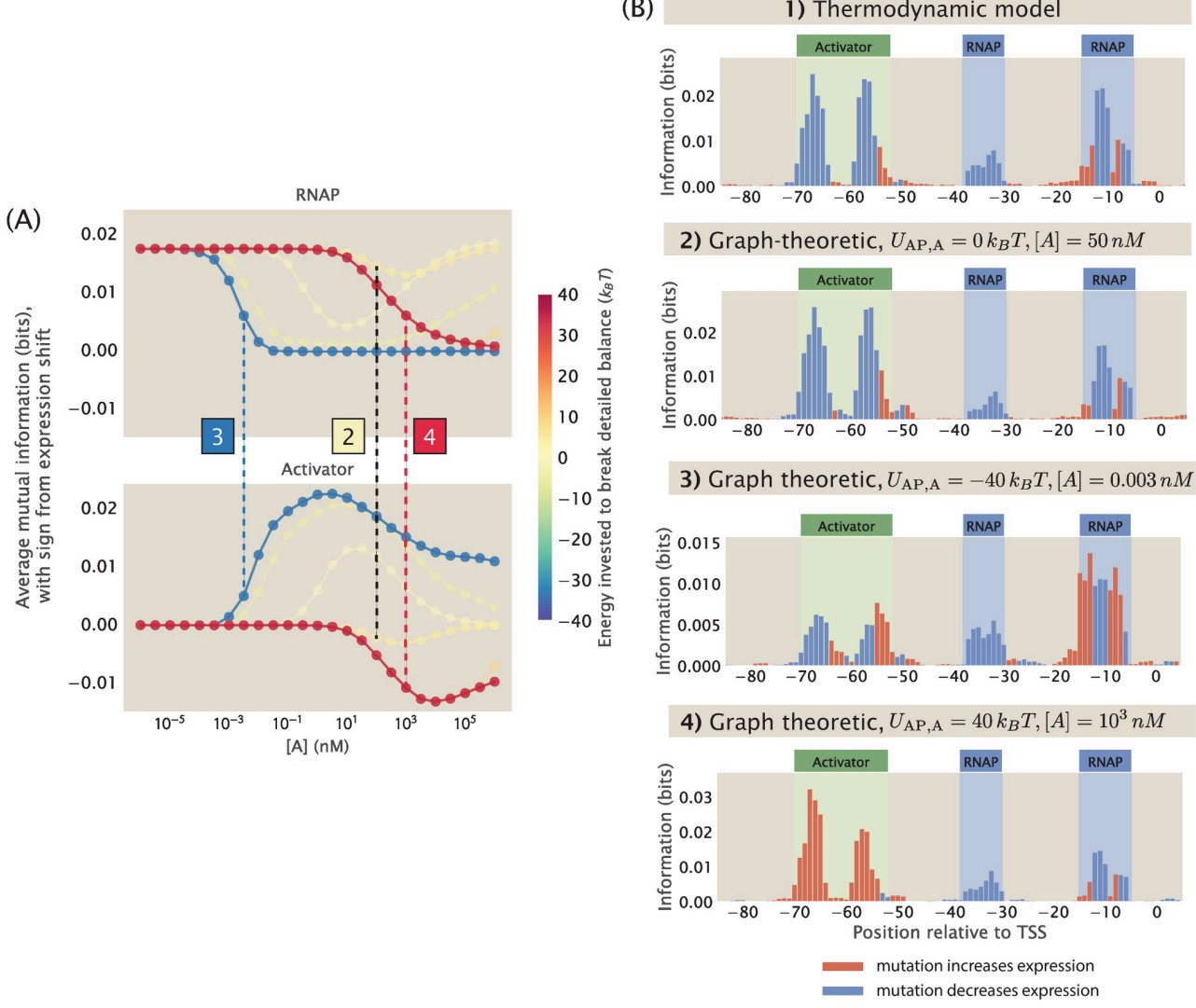

**Fig 15. Building synthetic datasets with broken detailed balance.** (A) Changes in average mutual information at the RNAP and activator binding sites when the concentration of the activator ($[A]$) and the energy invested to break the detailed balance at the AP $\rightarrow$ A edge ($U_{\mathrm{AP,A}}$, defined in S11 Appendix) are changed. The sign of the value on the y-axis is based on the expression shift values calculated using Eq 9. Each data point is the mean of average mutual information across 20 synthetic datasets with the corresponding $U_{\mathrm{AP,A}}$ and $[A]$. The numbered labels indicate datapoints for which the corresponding information footprints are shown in (B). (B) Information footprints built using the thermodynamic model and the graph-theoretic model. The second footprint is a graph-theoretic treatment of the equilibrium case.

3 of Fig 15B, under some conditions, the footprints are similar to what is obtained under equilibrium conditions. However, as shown in panel 4 of Fig 15B, in the cases where a positive energy is invested to break detailed balance at the AP → A edge and when activator concentration is high, the binding of activator displaces the binding of RNAP. As a result, the activator effectively behaves as a repressor. This generalized approach presents opportunities to explore MPRA datasets without being constrained by equilibrium assumptions.

## 3 Discussion

In this paper, we explore the landscape of sequence-energy-phenotype mapping by utilizing a new generation of sequence-specific thermodynamic models to simulate MPRAs. More generally, we use statistical mechanical models of gene expression to systematically explore the connection between mutations and level of gene expression. We have examined the effects of perturbing various experimental and biological parameters at multiple stages. Some parameters pertain to the initial library design, such as library size, mutation rate, and presence of mutational bias. Other parameters are built into the model itself, such as the copy number of the promoter and the transcription factors, which are parameters that may vary biologically or be affected by the design of experimental procedures.

We have demonstrated that our approach to simulate MPRA datasets has high flexibility and can easily be adapted to examine the effects of other perturbations not included in this paper. Furthermore, the computational nature of our approach allows full parameter searches to be done precisely and efficiently. For example, it would be both time-consuming and cost-prohibitive to experimentally determine the optimal library size and mutation rate as it would involve performing a large array of experimental tests. On the other hand, using our approach, we can efficiently build a series of synthetic datasets with different mutant libraries and determine the strategy for library design that is optimal for deciphering regulatory architectures.

Apart from informing the choice of experimental parameters, our work also helps to anticipate challenges involved in parsing information footprints. For example, in Sec 2.3.3, we predict how the signal in information footprints would be affected when there are overlapping binding sites. One potential usage of our computational pipeline is for building synthetic datasets that involve features that could lead to information footprints that are hard to parse. Since the synthetic datasets are built with prior knowledge of the underlying regulatory architectures, these datasets can be used to develop and improve algorithms for deciphering these architectures. This will increase confidence in the results when the same algorithms are used to analyze experimental datasets and determine the location of binding sites. Moreover, this will pave the way for automatically annotating binding sites for any given information footprint given MPRA data. To enable others who perform MPRAs in the context of transcriptional regulation to use our computational platform, we have made our code publicly available and are developing an interactive website where users may generate footprints given their own parameters of interest.

One limitation of our thermodynamic model is that since we rely on writing down states and weights models in order to predict the probability of transcriptionally active states, the combinatorial explosion can make it challenging for us to consider promoters that are regulated by three or more transcription factors. However, data from RegulonDB would suggest that over 80% of the promoters in *E. coli* fall under the six regulatory architectures that we discussed in Fig 4 [80]. Therefore, at least in prokaryotic genomes, we are confident that our sequence-specific thermodynamic models can be used to simulate MPRA datasets for the vast majority of promoters.

Furthermore, our current thermodynamic model only considers transcription initiation factors, while other types of transcription factors, such as elongation factors and termination factors, are also important for determining gene expression. It is challenging to include these factors in our model as it would involve additional kinetic terms that would have to be worked out, though we are extremely interested in developing these approaches as well in our future work. While this work does not directly address the challenge in understanding the role of transcription elongation factors and termination factors, we believe that by achieving a full understanding of transcription initiation factor binding through the efforts of both the computational and experimental MPRAs, it will help streamline strategies needed to decipher the roles of other types of transcription factors.

In addition, our thermodynamic model neglects the interaction between different genes in regulatory networks, which affects expression levels and may alter the expected signal in MPRA summary statistics such as information footprints. A future direction, therefore, involves building synthetic datasets of genetic networks. This would require an additional step where we modify the expression levels of each gene based on its dependency on other genes. This would not only improve the reliability of our prediction of expression levels, but these multi-gene synthetic datasets may also be used to test approaches for deciphering the architecture of regulatory networks.

Finally, while the vast amount of literature discussed in S1 Appendix gives us confidence in the validity of thermodynamic models, we acknowledge that there are many cases of transcriptional regulation in which detailed balance may be broken and thermodynamic models may no longer be appropriate. Our final results section (Sec 2.4) is a preliminary effort where we use graph-theoretic models of transcriptional regulation to produce synthetic datasets and summary statistics without enforcing equilibrium constraints. We expect that many more interesting and informative results can arise from non-equilibrium synthetic MPRA datasets. We are excited to further pursue this direction in our future work.

In summary, we have developed a theoretical framework for a widely used category of experiments in the field of transcriptional regulation. Our simulation platform establishes a systematic way of testing how well high-throughput methods such as MPRAs can be used to recover the ground truth of how the expression of a gene is transcriptionally regulated. This demonstrates the importance of developing theories of experiments in general, and we believe there is much untapped potential in extending similar types of theories to other areas of experimental work as well. Finally, we anticipate that this approach will also be useful in performing systematic studies on the relation between mutations in regulatory binding sites and the corresponding level of gene expression in a way that will shed light on both physiological and evolutionary adaptation.

## 4 Materials and methods

All code for generating synthetic datasets, information footprints, and expression shift matrices is written in Python. Please refer to Sec 2.1.1, S2 and S3 Appendices for details on our model and computational pipeline. All code for formal analysis and figure generation is available open source at https://github.com/RPGroup-PBoC/theoretical_regseq.

## Supporting information

**S1 Appendix. Models of the probabilities of microscopic states during transcription.** (PDF)

**S2 Appendix. Predicting the probability of RNA polymerase being bound using thermodynamic models.**
(PDF)

**S3 Appendix. Producing synthetic datasets from first principles.**
(PDF)

**S4 Appendix. Effects of mutation rate on information footprints.**
(PDF)

**S5 Appendix. Choosing an appropriate library size in MPRAs.**
(PDF)

**S6 Appendix. Changing transcription factor copy numbers under different regulatory logics.**
(PDF)

**S7 Appendix. Building thermodynamic models using the method of chemical potential.**
(PDF)

**S8 Appendix. Adding inducibility to thermodynamic models.**
(PDF)

**S9 Appendix. Noise from experimental procedures of MPRAs.**
(PDF)

**S10 Appendix. Modelling extrinsic noise in transcription initiation.**
(PDF)

**S11 Appendix. Calculating the probability of transcriptionally active states under non-equilibrium.**
(PDF)

## Acknowledgments

We would like to thank Justin Kinney, Sara Mahdavi, and Gabriel Salmon for helpful discussions and feedback on this manuscript.

## Author Contributions

**Conceptualization:** Rosalind Wenshan Pan, Tom Röschinger, Rob Phillips.

**Formal analysis:** Rosalind Wenshan Pan, Tom Röschinger.

**Funding acquisition:** Rob Phillips.

**Investigation:** Rosalind Wenshan Pan, Tom Röschinger, Kian Faizi.

**Methodology:** Rosalind Wenshan Pan, Tom Röschinger, Kian Faizi.

**Software:** Rosalind Wenshan Pan, Tom Röschinger, Kian Faizi.

**Supervision:** Rob Phillips.

**Visualization:** Rosalind Wenshan Pan.

**Writing – original draft:** Rosalind Wenshan Pan.

**Writing – review & editing:** Rosalind Wenshan Pan, Tom Röschinger, Kian Faizi, Hernan G. Garcia, Rob Phillips.

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
