## [Decision Letter · Decision Letter 0]

12 Apr 2024

Dear Prof. Phillips,

Thank you very much for submitting your manuscript "Dissecting endogeneous genetic circuits from first principles" for consideration at PLOS Computational Biology.

As with all papers reviewed by the journal, your manuscript was reviewed by members of the editorial board and by several independent reviewers. In light of the reviews (below this email), we would like to invite the resubmission of a significantly-revised version that takes into account the reviewers' comments.

Please address these concerns in particular:

(1) demonstrate the novelty of the results vs. previously published work from your lab and others

(2) underscore the limitations of the thermodynamic equilibrium approach

We cannot make any decision about publication until we have seen the revised manuscript and your response to the reviewers' comments. Your revised manuscript is also likely to be sent to reviewers for further evaluation.

Sincerely,

Alexandre V. Morozov, Ph.D.

Academic Editor

PLOS Computational Biology

Pedro Mendes

Section Editor

PLOS Computational Biology

Reviewer's Responses to Questions

**Comments to the Authors:**

Reviewer #1: Pan et al. have attempted to develop a theory for massively parallel reporter assays (MPRAs) that tries to delineate the relationship between DNA sequence and phenotype. In particular, the authors used the so-called “thermodynamic model” of gene regulation to develop to develop a pipeline to characterize how the different biological and experimental parameters control measured MPRA data. These parameters include the transcription factor (TF) binding site copy number, limited TF resource, etc. These biophysical parameters dictate the degree of mutual dependence between mutations in the regulatory region and expression levels. The authors systematically characterized the effects of various parameters on MPRA data. Moreover, authors showed how to optimise MPRA experimental designs.

The paper is very clearly written. The approach of “developing a theory of the experiment” is particularly important in biology. The paper should definitely be accepted. I have the following comments that the authors can try to address.

Main points

1) A figure describing the model and a brief introduction to the statistical ensemble / distinct microstates involved would be nice for people unfamiliar with the background literature. This can go to the methods or the supplementary section.

2) I wonder if the assumption of equilibrium is universally applicable to analyse MPRA data. Is it possible to extend the model to consider the scenario where the detailed balance condition is broken for TF binding to regulatory DNA.

Minor Points:

1) What is Ns? In line 79 it is referred to as the number of non-binding sites, in 82 it is referred to as the total number of base pairs in the genome.

2) The reference in line 76 (1) should be (17).

3) In line 322, it is mentioned that when R1>R2, the signal at the R1 binding site is higher but in the figure 8C 3) it is lower. I think the authors mean to say R2<r1.

4) In Figure 8C, subfigures 5) and 4) have the same captions.</r1.

Reviewer #2: See attached PDF with the review.

**Have the authors made all data and (if applicable) computational code underlying the findings in their manuscript fully available?**

Reviewer #1: Yes

Reviewer #2: None

PLOS authors have the option to publish the peer review history of their article (what does this mean?). If published, this will include your full peer review and any attached files.

Reviewer #1: No

Reviewer #2: No
---

## [Decision Letter · Decision Letter 1]

2 Oct 2024

Dear Prof. Phillips,

Thank you very much for submitting your manuscript "Deciphering regulatory architectures from synthetic single-cell expression patterns" for consideration at PLOS Computational Biology.

As with all papers reviewed by the journal, your manuscript was reviewed by members of the editorial board and by several independent reviewers. In light of the reviews (below this email), we would like to invite the resubmission of a significantly-revised version that takes into account the reviewers' comments.

Please address all the reviewer comments carefully, including their suggestions on improving the paper's flow and presentation and on making the computational pipeline more accessible. This will be the last round of major revisions for this manuscript.

We cannot make any decision about publication until we have seen the revised manuscript and your response to the reviewers' comments. Your revised manuscript is also likely to be sent to reviewers for further evaluation.

Sincerely,

Alexandre V. Morozov, Ph.D.

Academic Editor

PLOS Computational Biology

Pedro Mendes

Section Editor

PLOS Computational Biology

Reviewer's Responses to Questions

**Comments to the Authors:**

Reviewer #1: The authors have adequately addressed all the comments.

Reviewer #3: In this paper the authors provide thermodynamic models for 6 simple promoter architectures that are commonly found in E.coli: one repressor, one activator, two repressors act together, two repressors that have an xor relationship and an activator, a repressor and double activation. The binding energy is derived from the sequence motif and it is assumed that only binding of the TF at the original site can induce or repress transcription. Binding sites are assumed not to overlap, but new binding sites may appear in the genome and thus compete with the original site for limited TF-proteins. The model appears to require a number of detailed input parameters such as the number of RNA-polymerase and Transcription factor molecules in a cell (maybe a table with required input parameters would be nice to have).

Based on this the authors can then construct a synthetic RNA-seq data-set that predicts expression for all variants of the simulated promoter. This data is then used to calculate the mutual information between the sequences and the expression level, which can in a real life MPRA be used to predict the binding sites. The authors exemplify the LacZYA promoter. Here it would be nice to compare the simulated data to what the real data from their earlier papers show.

Until here this sounds like a general enough approach that might also work in eukaryotes, even though some of the assumptions, especially the lack of epistatic interactions and the strong position dependence, are unlikely to be correct.

Next, the authors go through experimental design aspects that I feel are rather specific to microbial systems: Specifically finding the optimal mutation rate to identify the TF footprints. Mutation rate and sequencing depth go hand in hand, and the authors also provide advice on that. However, what I am missing here is a discussion about how much this depends on biological properties: i.e. the size and information content of the TF motif as well as the expression level of the associated TF. Only in a later chapter it is mentioned that those factors are a cause for differences in the detection power, biasing the detection towards highly expressed TFs with strong binding sites.

Importantly, since the authors frame the paper as a computational pipeline for experimental design of MPRAs, I would appreciate that the simulation results are translated into the more traditional terms of power analysis such as the AUC.

As is the simulation results provoke interesting thoughts, but I do not see how a research can translate them into actionable advice on MPRA design and interpretation.

The insights with respect to promoter architecture and TF-concentration as well as competition for binding sites also remain not more than interesting thought experiments. Moreover, they are highly dependent on the initial assumptions, such as the importance of the exact location. Also here, I do not see concrete advice on how these insights will be helpful to real life examples with unknown TF architectures. Can the simulations help to distinguish such scenarios?

Adding the non-equilibrium model at the end is nice, but again I don’t quite see the practical implications. If I understand correctly, only the simplest architecture is implemented and for some conditions the simpler and more flexible equilibrium model will suffice. Again a comparison of simulations to real life data would be helpful, to guide the user towards what model is appropriate.

Furthermore, the paper is too long, it should be possible to shorten. Each chapter centres on a specific scenario and then repeats a similar train of logic. It should be possible to shorten and only describe the implications.

Most importantly, the computational pipeline that I found on GitHub is a collection of jupyter notebooks that is rather sparsely annotated.

If this is to be of use to other people more work is needed here. At an absolute minimum the functions provided in tregseq require a detailed description including I/O.

In summary, given the limitations of the model both the introductory remarks and the discussion are vastly overstated. This paper could be of merit if a few results are clearly formulated and not buried between lengthy derivations and the overstated introduction and discussion. For most parts, this paper reads more like a review than the description of a computational pipeline as is claimed in the discussion. The authors need to decide on a focus and rewrite accordingly.

Minor comments

I find the title misleading. Even though a single cell is simulated, it is not intended for single cell MPRA analysis. Also the title should make it clear that the focus is on microbial systems not to mislead people like me who work on mammalian single cell genomics.

Many of the appendices appear to be basic repetitions of what is already in the main paper, this makes it extremely difficult to fin the relevant information. Again I could not find a real pipeline description.

**Have the authors made all data and (if applicable) computational code underlying the findings in their manuscript fully available?**

Reviewer #1: None

Reviewer #3: Yes

PLOS authors have the option to publish the peer review history of their article (what does this mean?). If published, this will include your full peer review and any attached files.

Reviewer #1: No

Reviewer #3: No
---

## [Decision Letter · Decision Letter 2]

4 Dec 2024

Dear Prof. Phillips,

We are pleased to inform you that your manuscript 'Deciphering regulatory architectures of bacterial promoters from synthetic expression patterns' has been provisionally accepted for publication in PLOS Computational Biology.

Best regards,

Alexandre V. Morozov, Ph.D.

Academic Editor

PLOS Computational Biology

Pedro Mendes

Section Editor

PLOS Computational Biology

Feilim Mac Gabhann

Editor-in-Chief

PLOS Computational Biology

Jason Papin

Editor-in-Chief

PLOS Computational Biology

Reviewer's Responses to Questions

**Comments to the Authors:**

Reviewer #3: Honestly, I found that the long responses to my first review rather confused me more than they clarified. The authors also repeatedly implied that the issue was my limited knowledge of the field. This might well be, I am not expert in thermodynamic modelling. I know about MPRAs from an application side and for somebody like me I don't see any application for the presented models. I don't have the time and the energy to go again through this extremely long and complex paper to argue point by point. Therefore I refrain from giving another recommendation.

**Have the authors made all data and (if applicable) computational code underlying the findings in their manuscript fully available?**

Reviewer #3: Yes

PLOS authors have the option to publish the peer review history of their article (what does this mean?). If published, this will include your full peer review and any attached files.

Reviewer #3: No

---

## [Editor Report · Acceptance letter]

18 Dec 2024

PCOMPBIOL-D-24-00164R2 

Deciphering regulatory architectures of bacterial promoters from synthetic expression patterns

Dear Dr Phillips,

I am pleased to inform you that your manuscript has been formally accepted for publication in PLOS Computational Biology. Your manuscript is now with our production department and you will be notified of the publication date in due course.

With kind regards,

Anita Estes
